# Visual Geometry Transformer in the Wild: Distractor-Free 3D Reconstruction

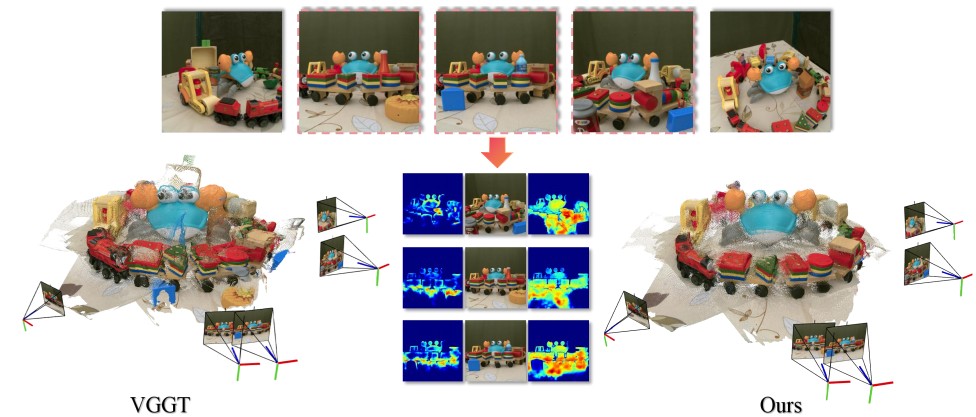

Figure 1: Our VGTW is a feed-forward method for distractor-free 3D reconstruction from inconsistent input images. The middle panel visualizes confidence maps from both models, highlighting regions of reliable prediction.

## Abstract

Current end-to-end multi-view 3D reconstruction methods achieve impressive results, but rely on a restrictive static assumption: the scenes is entire distractor-free with perfect cross-view geometry. This reliance on idealized inputs causes even the most advanced methods to fail in real-world settings, where transient distractors and occlusions present. To address this, we propose *Visual Geometry Transformer in the Wild* (VGTW), an end-to-end framework for robust reconstruction from inconsistent views. At its core, we isolate and suppress distractor-affected regions while preserving the consistent components across views. Specifically, we introduce a Distractor-aware Training (DAT) strategy that separates clean features from distractor-contaminated ones in the attention mechanism while enforcing feature consistency across images. To enable this, we train the model with an auxiliary mask prediction head, using supervision from a new dataset we collected with pixel-level distractor masks. The resulting VGTW model is a feed-forward network that directly outputs clean, distractor-free point clouds. Remarkably, it requires no additional 3D supervision, remains computationally efficient, and is compatible with existing pipelines. Extensive experiments validate our approach, demonstrating state-of-the-art performance and robust generalization in diverse, real-world scenarios.

## 1 Introduction

Multi-view 3D reconstruction is a cornerstone of modern computer vision. Traditionally, it has relied on complex, modular pipelines involving stages such as feature extraction, matching, and triangulation. However, recent paradigms shift treats this challenge as a direct end-to-end regression problem Wang et al. (2024; 2025a). By training on large-scale datasets, those models learn to directly estimate 3D information, like camera parameters and point maps, from the input images.

In this way, it avoids compounding errors and provides a simpler, more robust solution. Yet, this approach, despite all its promise, rests on a fragile assumption: that the scene is entirely static with dense visual correspondence. In other words, they expect every point in the scene to remain consistent across all input images. Unfortunately, this assumption often fails in real-world scenarios.

In contrast, real-world captures frequently contain transient distractors such as moving people or vehicles. Those elements disrupt inter-frame consistency, making it difficult to align static components of the scene and ultimately degrading reconstruction quality. While VGGT Wang et al. (2025a) already attempts to address this by predicting pixel-wise confidence and filtering out low-confidence regions, the method remains brittle. As illustrated in the left part of Fig. 1, simply thresholding confidence scores often fails to reliably remove distractors without also discarding valuable scene information.

While the challenge of transient distractors is not new, existing solutions are poorly suited for this end-to-end paradigm. Prior methods based on NeRF Martin-Brualla et al. (2021); Ren et al. (2024) and 3D Gaussian Splatting Kerbl et al. (2023); Kulhanek et al. (2024), for example, rely on slow, per-scene iterative optimization to model transient elements, often by analyzing rendering residuals. Those approach are not only time-consuming, but also fundamentally incompatible with newer models that operate without explicit optimization. This disconnect highlights a critical gap for a feed-forward method that can produce a clean point cloud from inconsistent views in a single forward pass.

To fill this gap, we introduce the Visual Geometry Transformer in the Wild (VGTW), an end-to-end framework for robust 3D reconstruction from real-world, inconsistent image collections. Our method is grounded in a simple yet powerful principle: we explicitly isolate and suppress distractor-affected regions while enforcing geometric consistency across stable scene components.

Specifically, we introduce a Distractor-aware Training (DAT), a strategy that fine-tunes the attention mechanism via Low-Rank Adaptation (LoRA) to separate clean features from those contaminated by transient objects. This is guided by two novel objectives: a *Distractor Suppression Loss* to penalize the influence of distractors, and a *Cross-View Consistency Loss* to reinforce the consistency of geometric features.

The key to enabling this separation is our training strategy. We train VGTW with an auxiliary mask prediction head on RobustNeRF-Mask dataset, a new dataset we collected with pixel-level distractor annotations. Crucially, because all supervision is 2D, the model learns to identify distractors without needing any 3D ground truth labels, which makes the training highly efficient.

The resulting VGTW model is a feed-forward network that directly outputs a clean, distractor-free 3D geometry and cameras. Remarkably, despite being trained on our relatively small dataset, VGTW learns a robust and generalizable understanding of distractors, performing robustly on a wide range of unseen scenarios.

The contribution of this work could be summarized as:

- We introduce VGTW, the first end-to-end, feed-forward framework that robustly reconstructs 3D scenes from in-the-wild images containing transient distractors.
- We propose a novel Distractor-aware Training strategy to explicitly suppress distractor features scene elements without 3D supervision.
- We introduce the RobustNeRF-Mask dataset, a new public resource with multi-view images and pixel-perfect distractor masks to train and evaluate our model.
- VGTW achieves state-of-the-art (SOTA) reconstruction quality on a wild range of benchmark, generalized to diverse and unseen real-world scenarios, and outperforming existing methods in robustness and reconstruction fidelity.

## 2 RELATED WORKS

### 2.1 3D RECONSTRUCTION IN-THE-WILD

3D reconstruction in-the-wild aims to recover scene geometry and appearance from unconstrained images, like casual snapshots or internet photos, often plagued by dynamic distractors, varying

lighting, and occlusions. Early NeRF extensions Martin-Brualla et al. (2021); Ren et al. (2024); Du et al. (2021); Gao et al. (2021); Li et al. (2021; 2022); Wang et al. (2021); Wu et al. (2022); Chen et al. (2022) addressed these issues; for example, NeRF-W Martin-Brualla et al. (2021) models variable appearance through per-image latent embeddings and handles transient occluders via a separate volumetric field with uncertainty estimation. Recent advancements like NeRF On-the-go Ren et al. (2024) exploit predicted uncertainty to robustly eliminate distractors even in high-occlusion scenarios, enabling faster convergence on casually captured sequences. With the emergence of 3D Gaussian Splatting (3DGS), methods Kulhanek et al. (2024); Sabour et al. (2025); Xu et al. (2024); Zhang et al. (2024a); Dahmani et al. (2024) have shifted toward explicit representations for efficiency. WildGaussians Kulhanek et al. (2024) integrates robust DINO features and an appearance modeling module to address occlusions and photometric variations in uncontrolled settings. SpotLessSplats Sabour et al. (2025) employs pre-trained features and robust optimization to ignore transient distractors during reconstruction. Wild-GS Xu et al. (2024) aligns pixel appearance features to local Gaussians through triplane sampling from reference images, facilitating efficient restoration of high-frequency details. However, prior NeRF and 3DGS-based methods rely on slow iterative optimization, making them incompatible with feed-forward approaches.

## 2.2 FEED-FORWARD 3D RECONSTRUCTION

Feed-forward models Wang et al. (2024); Leroy et al. (2024); Cabon et al. (2025;?); Zhang et al. (2025); Wang & Agapito (2024); Wang et al. (2025b) have recently emerged as an alternative to traditional optimization-based pipelines for 3D scene reconstruction, enabling direct regression of geometric structures from images in a single pass. Pioneering works such as DUSt3R Wang et al. (2024) process image pairs to predict dense pointmaps aligned to a reference camera frame, supporting unconstrained reconstruction from arbitrary views. Follow-up methods like MASt3R Leroy et al. (2024) incorporate dense local features and matching supervision for enhanced robustness, while MUSt3R Cabon et al. (2025) and MV-DUSt3R+ Tang et al. (2025) extend the pairwise paradigm to small multi-view sets, albeit with challenges in alignment for larger scenes. In order to further handle the dynamic scenarios reconstruction, methods like MonST3R Zhang et al. (2024b) incorporate optical flow to model motion, while Easi3R Chen et al. (2025) refines dynamic segmentation through attention re-weighting.

Later developments emphasize simultaneous handling of multiple frames to address scalability issues. For example, CUT3R Wang et al. (2025b) and Spann3R Wang & Agapito (2024) leverage stateful recurrent architectures or spatial memory for online dense reconstruction from image streams or extensive collections. FLARE Zhang et al. (2025) decomposes the task into sequential pose estimation and geometry prediction to improve efficiency in sparse-view settings. Scalable methods include Fast3R Yang et al. (2025), which reconstructs from thousands of unordered images in one forward pass without post-alignment; VGGT Wang et al. (2025a), which regresses full 3D attributes such as poses, pointmaps, and depths across multi-view inputs; and $\pi^3$ Wang et al. (2025c), which employs a permutation-equivariant network for geometry learning without a fixed reference.

These approaches, however, assume static scenes with reliable dense correspondences and exhibit limited robustness to transient distractors in dynamic inputs. In contrast, our VGTW introduces Distractor-aware Training strategy and a dedicated decoder to mitigate feature pollution from such elements while effectively filtering them out.

## 3 PRELIMINARY

### 3.1 FEED-FORWARD MULTI-VIEW 3D RECONSTRUCTION

In this work, we build upon the frameworks established by VGGT Wang et al. (2025a) and $\pi^3$ Wang et al. (2025c), two state-of-the-art approaches for feed-forward multi-view 3D reconstruction. The input consists of a sequence of $N$ RGB images $(I_i)_{i=1}^{N}$, where each $I_i \in \mathbb{R}^{3 \times H \times W}$ captures the same 3D scene from different perspectives. Both VGGT and $\pi^3$ map this sequence to a set of 3D annotations, one per frame, as defined by the following functions:

$$f_{\text{VGGT}}\left((I_i)_{i=1}^{N}\right) = (g_i, D_i, P_i, C_i, T_i)_{i=1}^{N}, \tag{1}$$

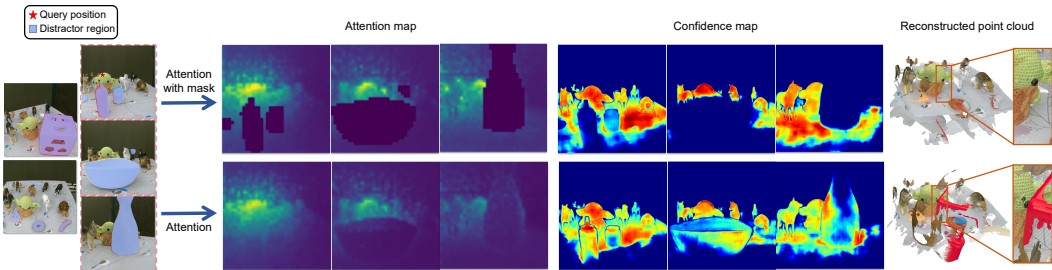

Figure 2: The difference of using mask in attention shown in attention map, generated confidence map and reconstructed point cloud.

$$f_{\pi^3}\left((I_i)_{i=1}^N\right) = (g_i, D_i, P_i, C_i)_{i=1}^N, \tag{2}$$

where $g_i \in \mathbb{R}^9$ represents the camera parameters (intrinsics and extrinsics) for image $I_i$, $D_i \in \mathbb{R}^{H \times W}$ denotes the depth map, and $P_i \in \mathbb{R}^{3 \times H \times W}$ corresponds to the point map, and $C_i$ is corresponding confidence of dense prediction. For VGGT, an additional output $T_i \in \mathbb{R}^{C \times H \times W}$ captures a grid of $C$-dimensional features for point tracking. These dense outputs are generated by their respective DPT Ranftl et al. (2021) heads in VGGT, while $\pi^3$ employs a permutation-equivariant architecture to ensure robustness to input view order. This study utilizes $g_i$ and $D_i$ from either VGGT or $\pi^3$ to compute the point cloud $P_i$ and the intermediate feature $F_d$ from the depth prediction head.

### 3.2 IN-THE-WILD 3D RECONSTRUCTION: CHALLENGES AND INSIGHTS

**Problem statement.** In this paper, we target the problem of robust 3D reconstruction of *static scene geometry* from unstructured, in-the-wild image sequences. By *unstructured*, we mean that the image contains transient distractors that are not consistent across all views. This may include dynamic objects, occluders, and non-rigid deformations.

In particular, distractors hinder reliable matching of features extracted from image pairs. This interference manifests as geometric errors during view warping: a pixel $\mathbf{p}$ in image $I_i$ is back-projected to a 3D point using its depth $D_i(\mathbf{p})$ and camera pose $g_i$, then re-projected to image $I_j$ via the relative pose $\Delta g_{ij}$, yielding a projected pixel $\mathbf{p}'$ and depth $d'$. Distractors result in high depth residuals $|d' - D_j(\mathbf{p}')|$, signaling inconsistencies.

Unlike prior methods that rely on per-scene optimization, we want to learn a single, feed-forward model that directly maps the set of inconsistent views $(I_i)_{i=1}^N$ to the 3D geometry representing the underlying static scene.

Unfortunately, this is particularly difficult, because the presence of such distractors breaks the static world assumption that classical multi-view geometry relies on. Conventional methods, which depend on finding stable corresponding points for triangulation, become unreliable when these geometric inconsistencies arise. While Easi3R Chen et al. (2025) introduces a training-free pairwise processing method detecting attention to tokens violating epipolar constraints (e.g., from motion) via aggregated cross-attention maps. However, this pairwise focus limits cross-attention to only two views, missing multi-view interactions essential for global consistency in unordered, in-the-wild image sets with diverse viewpoints. Its reliance on low inter-frame correlations for detection often misclassifies static regions as dynamic under large viewpoint shifts. As a result, the reconstruction problem becomes ill-posed, often yielding incomplete, noisy, or entirely incorrect geometry.

**Problem analysis: Attention Is Sufficient to Remove Distractors.** To understand how distractors degrade feed-forward 3D reconstruction, we examine two representative models, VGGT and $\pi^3$, which rely on attention mechanisms to aggregate information across views.

Experiment Setup. We probe the models using a controlled analysis. For a static scene region (marked by a red query point), we visualize (1) the attention map, (2) the confidence map, and

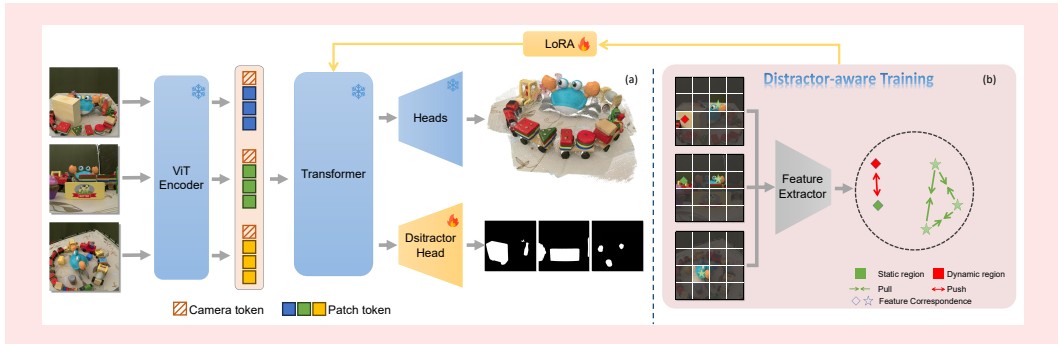

Figure 3: Overall pipeline of VGTW. (a) VGTW architecture for 3D reconstruction in the wild; (b) Distractor-aware Training (DAT) strategy to suppress distractors and enhance feature robustness.

(3) the reconstructed point cloud (see Fig. 2). We assume access to *ground-truth distractor masks* that identify distractor regions. This allows us to test whether explicitly suppressing attention to these regions improves reconstruction quality.

Observations 1: Attention leaks to distractors. When querying a static region (red), standard attention assigns high weights to distractor regions (blue). This causes the model to treat distractor content as reliable, inflating confidence scores for spurious points. As a result, these false points survive confidence-based filtering and appear in the final reconstruction.

Observations 2: Masking attention removes distractors. If we prevent the model from attending to known distractors, everything improves. To do this, we zero out attention to known distractor regions by setting their logits to $-\infty$. This redirects focus to consistent static correspondences across views. As shown in Fig. 2, this leads to a sharp drop in distractor confidence, effective removal of spurious points, and a cleaner, more complete point cloud.

This shows that just by cleaning up attention without changing anything else, we can effectively remove distractors. The model then better uses geometric consistency across views, leading to more accurate and reliable 3D reconstructions. Of course, ground-truth masks are not available at test time. To make this practical, we later train the model to *learn* to suppress such distracting attention patterns automatically.

## 4 METHOD

Build on top of the above analysis, we present **Visual Geometry Transformer in the Wild** (VGTW), a feedforward model that achieve robust 3D multi-view reconstruction from inconsistent views. The core principle of VGTW is to explicitly isolate and suppress regions affected by distractors. This design allows VGTW to flexibly build upon existing frameworks like VGGT and $\pi^3$ while significantly improving robustness in real-world, uncontrolled settings.

**Overview.** The pipeline of VGTW is shown in Fig.3(a). Similar to prior works Wang et al. (2025a;c), it takes a sequence of $N$ RGB images $(I_i)_{i=1}^N$ as input and embed them into patch tokens using pretrained DINO encoders Oquab et al. (2023). These tokens are subsequently processed through a series of interleaved view-wise and global self-attention layers to produce the feature $(H_i)_{i=1}^N$. Similar to VGGT and $\pi^3$, we map the learned feature $(H_i)_{i=1}^N$ to an output tuple of 3D reconstruction $(g_i, D_i, P_i, T_i)_{i=1}^N$ using a decoder with parameters $\theta^*$:

$$(g_i, D_i, P_i, T_i) = \text{Heads}(H_i, \theta^*). \tag{3}$$

While VGTW preserves the same core architecture, it introduces two key improvement. (1) We finetune the model using a distractor suppression loss and a cross-view consistency loss in Sec. 4.1 to improve geometric consistency. (2) We add a mask head that predicts distractor masks (Sec. 4.2), which are used to filter the output distractor. The overall training objective is detailed in Sec. 4.3.

Notably, training VGTW *requires no 3D annotations*; its training requires solely on RGB images and corresponding distractor masks, making it extremely lightweight and practical.

Figure 4: The difference of using VGTW shown in attention map, feature similarity map, generated confidence map and reconstructed point cloud.

### 4.1 DISTRACTOR-AWARE TRAINING

Inspired by the analysis in Sec. 3.2 and visualized in Fig. 2, masking distractors within the attention mechanism effectively suppresses their influence, leading to cleaner point clouds by reducing spurious confidence scores. To address this, we propose a novel *Distractor-aware Training* strategy that fine-tunes the pre-trained view-wise and global self-attention layers in the transformer backbone. Specifically, we employ Low-Rank Adaptation (LoRA) Hu et al. (2022) for this fine-tuning, enabling the model to selectively diminish salience assigned to dynamic regions while promoting focused learning of consensual geometric information across frames. This approach preserves the efficiency of feed-forward inference and enhances generalization to in-the-wild scenarios.

To promote the separation of clean, static features from distractor-contaminated ones, we introduce two novel supervisory signals during finetuning: the Distractor Suppression Loss and the Cross-View Consistency Loss, as illustrated in Fig. 3(b). These losses operate on the intermediate features $H_i$ output by the attention layers, leveraging feature correspondences to enforce robustness against transients.

**Distractor Suppression Loss.** Post-attention features encode rich 3D geometric cues, where high pairwise similarities often indicate correspondences across views. Transient distractors disrupt this consistency and should show low similarities to patches in other views. To isolate them, we push distractor features away from their most similar neighbors in alternate views, while leaving static features unchanged. Formally, for each feature $h_i \in H_i$ in view $i$, we identify its most similar counterpart $\tilde{h}i = \arg\max h \in H_{j \neq i} S(h_i, h)$ across other views $j \neq i$, where $S(h, \tilde{h}) = \frac{h \cdot \tilde{h}}{|h||\tilde{h}|}$ is the similarity function. Using the predicted mask $M_i$ to classify regions, the loss is defined as:

$$\mathcal{L}_{\text{supp}} = \sum_{i=1}^{N} \sum_{h \in H_i} M_i(h) \cdot \max\left(0, S(h, \tilde{h}) - \gamma_d\right),$$

where $\gamma_d$ is a small margin hyperparameter to encourage low similarity, and the summation applies only to distractor regions ($M_i(h) = 1$). This hinge-like loss mitigates the polluting effect of distractors on cross-view reasoning by keeping their similarities below the margin.

**Cross-view Consistency Loss.** To complement the distractor suppression, we enforce consistency among static regions across views by pulling together patch features that represent the same scene areas, adopting an approach akin to spatial feature clustering. Specifically, for each static feature $h_i \in H_i$, we identify its most similar static neighbor in each other view $j \neq i$ as $\tilde{h}i, j = \arg\max h \in H_j S(h_i, h)$. Crucially, to avoid imposing spurious alignments when reliable correspondences are absent, the extent of pulling for each pair is modulated by their similarity $S(h_i, \tilde{h}_{i,j})$, ensuring stronger optimization for high-similarity pairs and weaker for low-similarity ones. Formally, the loss is defined as:

$$\mathcal{L}_{\text{cons}} = \sum_{i=1}^{N} \sum_{h \in H_i} (1 - M_i(h)) \cdot \sum_{j \neq i} S(h_i, \tilde{h}_{i,j}) \cdot \max\left(0, \gamma_s - S(h_i, \tilde{h}_{i,j})\right),$$

where $\gamma_s$ is a margin hyperparameter that encourages similarities to exceed this threshold.

As shown in Fig. 4, the combination of $\mathcal{L}$supp and $\mathcal{L}$cons effectively mitigates the impact of distractors in the attention mechanism, separating clean features from contaminated ones. The figure's

feature distance maps depict the average similarity of each feature to its most similar neighbors in alternate views. As evident, distractor similarities decrease, indicating successful isolation from static regions.

## 4.2 DISTRACTOR HEAD.

As discussed in Sec. 3.2, distractors tend to capture salience from the attention mechanism, resulting in higher confidence levels for distractors and preventing them from being filtered out. This phenomenon is also illustrated in Fig. 2. Furthermore, since the proportion of distractors varies across different scenarios, a predefined threshold may fail to completely eliminate them while risking the loss of valuable information. To address this, we design a distractor head that outputs binary masks $(M_i)_{i=1}^N$ to indicate which regions are distractors:

$$M_i = \text{Head}_{\text{mask}}(H, \theta),\tag{4}$$

where $\theta$ are the trainable parameters of the distractor head. To train the distractor head, we introduce supervision from the ground-truth mask $\hat{M}_i$. The training loss for the distractor head is defined using the binary cross-entropy (BCE) loss:

$$\mathcal{L}_{\text{mask}} = \sum_{i=1}^N \sum_{k=1}^K \left[ \hat{M}_i^{(k)} \log M_i^{(k)} + (1 - \hat{M}_i^{(k)}) \log(1 - M_i^{(k)}) \right],\tag{5}$$

where $K$ is the number of pixels in each mask. Building on this, we can further filter the point cloud using the prediected mask $M_i$, allowing for more precise removal of distractors.

## 4.3 MODEL TRAINING

The overall training objective combines these with the mask loss: $\mathcal{L} = \lambda_1 \mathcal{L}_{\text{mask}} + \lambda_2 \mathcal{L}_{\text{supp}} + \lambda_3 \mathcal{L}_{\text{cons}}$, where $\lambda_1$ and $\lambda_2$ are weighting hyperparameters. By integrating Distractor-aware Training with these targeted losses, attention mechanism strengthens cross-view alignment for reliable static features, enhancing robustness in dynamic environments without requiring explicit 3D supervision. By testing on the various datasets VGTW achieves state-of-the-art distractor handling in a fully feed-forward manner, and show great generalization ability in zero-shot test, marking a key novelty over prior iterative optimization methods.

## 5 EXPERIMENT

### 5.1 IMPLEMENTATION DETAILS

**Dataset preparation.** We train VGTW using images and paired distractor masks from our newly annotated RobustNeRF-Mask dataset and the NeRFOSR dataset Rudnev et al. (2022). RobustNeRF-Mask extends the original RobustNeRF dataset by using SAM2 Ravi et al. (2024) to label distractor masks for each frame, providing about 1000 annotated images and addressing the absence of such labels. The combined datasets provide thousands of training images, covering diverse dynamic scenes, including indoor and outdoor with transient distractors, without relying on 3D supervision like depth or poses.

**Training.** Following practices in VGGT and $\pi^3$, we initialize from pretrained VGGT weights and finetune . Training uses AdamW optimizer with a cosine learning rate scheduler for 50 epochs. We employ bfloat16 precision, gradient checkpointing to improve GPU memory and computational efficiency. Key hyperparameters include the confidence threshold $\tau = 0.5$ for point filtering, similarity margin $\gamma = 0.1$ in the distractor suppression loss, mask loss weight $\lambda_{\text{mask}} = 1.0$, and overall loss weights $\lambda_1 = 0.2$, $\lambda_2 = 0.01$, $\lambda_3 = 0.05$.

**Baseline.** We compare our VGTW(VGGT) and VGTW($\pi^3$) variants against state-of-the-art methods, including DUSt3R Wang et al. (2024) and MASt3R Leroy et al. (2024) (pair-wise processing with pointmap regression and global alignment), and VGGT Wang et al. (2025a) (primary backbone), $\pi^3$ Wang et al. (2025c), and Fast3R Yang et al. (2025) (multi-frame simultaneous processing in a single forward pass). Evaluations focus on point map estimation and depth estimation tasks on the RobustNeRF dataset Sabour et al. (2023) and the unseen NeRF-on-the-go dataset Ren et al. (2024).

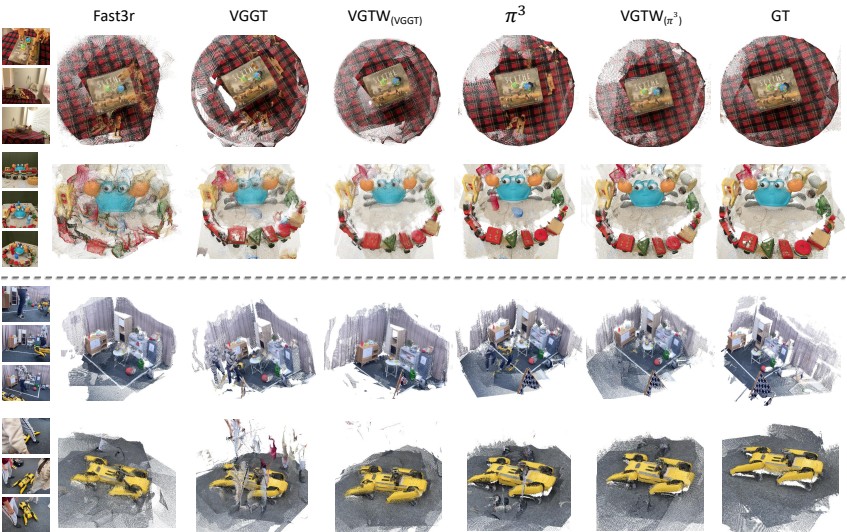

Figure 5: Qualitative results of point map estimation. Above the dashed line: performance on the RobustNeRF dataset. Below: performance on the NeRF On-the-go dataset.

## 5.2 POINT MAP ESTIMATION

Table 1: Quantitative comparisons of 3D point map estimation performance on NeRF on-the-go dataset Ren et al. (2024).

| | Low Occlusion | | | | | | Medium Occlusion | | | | | | High Occlusion | | | | | | Average | | |
| | Mountain | | | Fountain | | | Corner | | | Patio | | | Spot | | | Patio-High | | | | | |
| | Acc↓ | Comp↓ | NC↑ | Acc↓ | Comp↓ | NC↑ | Acc↓ | Comp↓ | NC↑ | Acc↓ | Comp↓ | NC↑ | Acc↓ | Comp↓ | NC↑ | Acc↓ | Comp↓ | NC↑ | Acc↓ | Comp↓ | NC↑ |
|---|---|---|---|---|---|---|---|---|---|---|---|---|---|---|---|---|---|---|---|---|---|
| DUSt3R | 0.015 | **0.012** | 0.690 | 0.070 | 0.059 | 0.674 | 0.046 | 0.050 | **0.820** | 0.042 | 0.154 | 0.706 | 0.017 | 0.061 | **0.831** | 0.034 | 0.143 | 0.764 | 0.037 | 0.080 | **0.747** |
| MaSt3R | 0.019 | 0.057 | 0.645 | 0.069 | 0.184 | 0.627 | 0.058 | 0.087 | 0.754 | 0.069 | 0.184 | 0.627 | 0.025 | 0.184 | 0.778 | 0.030 | 0.248 | 0.724 | 0.045 | 0.157 | 0.692 |
| Fast3R | 0.017 | 0.019 | 0.686 | 0.057 | 0.079 | 0.592 | 0.056 | 0.045 | 0.771 | 0.057 | **0.079** | 0.592 | 0.016 | 0.073 | 0.770 | 0.043 | **0.118** | 0.669 | 0.041 | 0.069 | 0.680 |
| VGGT | 0.019 | 0.024 | 0.696 | 0.060 | 0.090 | 0.630 | 0.059 | 0.076 | 0.630 | 0.043 | 0.164 | 0.633 | 0.027 | 0.139 | 0.663 | 0.037 | 0.384 | 0.586 | 0.041 | 0.146 | 0.640 |
| VGTW$_{VGGT}$ | 0.008 | 0.015 | **0.738** | 0.057 | **0.042** | 0.652 | 0.040 | 0.048 | 0.740 | **0.041** | 0.141 | **0.707** | **0.016** | 0.050 | 0.755 | 0.033 | 0.406 | 0.631 | 0.033 | 0.117 | 0.704 |
| $\pi^3$ | 0.012 | 0.023 | 0.707 | 0.055 | 0.078 | 0.691 | 0.039 | 0.056 | 0.762 | 0.128 | 0.135 | 0.589 | 0.024 | 0.037 | 0.772 | 0.047 | 0.118 | 0.734 | 0.051 | 0.074 | 0.709 |
| VGTW($\pi^3$) | **0.005** | 0.023 | 0.551 | **0.052** | 0.043 | **0.699** | **0.019** | **0.032** | 0.762 | 0.050 | 0.114 | 0.668 | 0.018 | **0.032** | 0.693 | **0.020** | 0.119 | **0.776** | **0.027** | **0.060** | 0.692 |

Table 2: Quantitative comparisons of 3D point map estimation performance on RobustNerf dataset Sabour et al. (2023).

| | Android | | | Crab1 | | | Crab2 | | | Yoda | | | Average | | |
| | Acc↓ | Comp↓ | NC↑ | Acc↓ | Comp↓ | NC↑ | Acc↓ | Comp↓ | NC↑ | Acc↓ | Comp↓ | NC↑ | Acc↓ | Comp↓ | NC↑ |
|---|---|---|---|---|---|---|---|---|---|---|---|---|---|---|---|
| DUSt3R | 0.054 | 0.141 | 0.699 | 0.023 | 0.041 | 0.653 | 0.028 | 0.054 | 0.659 | 0.018 | 0.035 | 0.774 | 0.031 | 0.068 | 0.696 |
| MaSt3R | 0.052 | 0.294 | 0.696 | 0.026 | 0.403 | 0.565 | 0.037 | 0.327 | 0.578 | 0.037 | 0.526 | 0.623 | 0.038 | 0.388 | 0.616 |
| Fast3R | 0.029 | 0.134 | 0.693 | 0.041 | 0.117 | 0.596 | 0.039 | 0.106 | 0.624 | 0.027 | 0.062 | 0.728 | 0.034 | 0.105 | 0.660 |
| VGGT | 0.027 | 0.033 | 0.746 | 0.020 | 0.046 | 0.633 | 0.017 | 0.046 | 0.662 | 0.018 | 0.055 | 0.694 | 0.021 | 0.045 | 0.684 |
| VGTW(VGGT) | 0.007 | 0.017 | **0.791** | 0.017 | 0.035 | 0.666 | **0.009** | 0.020 | 0.739 | 0.012 | 0.029 | 0.764 | 0.011 | 0.025 | 0.740 |
| $\pi^3$ | 0.015 | 0.018 | 0.750 | 0.021 | 0.019 | **0.723** | 0.015 | 0.012 | **0.743** | 0.015 | 0.016 | **0.799** | 0.017 | 0.016 | **0.754** |
| VGTW($\pi^3$) | **0.005** | **0.007** | 0.780 | **0.010** | **0.011** | 0.685 | 0.013 | **0.011** | 0.688 | **0.011** | **0.010** | 0.718 | **0.010** | **0.010** | 0.718 |

We evaluate point map estimation on the NeRF-on-the-go and RobustNeRF datasets, featuring dynamic scenes with transient distractors. Since these datasets lack clean point clouds, we generate ground-truth using pretrained $\pi^3$ model on distractor-free images from the same scenes. For each evaluation, we randomly sample 10 images, potentially mixing clean and distractor-containing ones. Metrics include accuracy (Acc ↓), completeness (Comp ↓), and normal consistency (NC ↑), computed after Umeyama alignment of predicted point clouds to ground truth.

Table 1 shows results on the unseen NeRF-on-the-go dataset. Our VGTW(VGGT) and VGTW($\pi^3$) variants achieve state-of-the-art performance, outperforming baselines across low, medium, and high occlusion levels with notable gains in Acc and Comp under high occlusion—demonstrating superior distractor handling—and exhibit substantial improvements over their original counterparts, VGGT and $\pi^3$, respectively. On RobustNeRF, Table 2 highlights our VGTW($\pi^3$) yielding the lowest Acc and Comp in most scenes—underscoring effective suppression of transients without 3D labels—and showing significant enhancements over original VGGT and $\pi^3$.

Qualitatively, as illustrated in Fig. 5, our VGTW produce cleaner point clouds, effectively removing transients while promoting better alignment of static regions, like the crab's structure in the second row—compared to baselines like Fast3R and VGGT, which retain artifacts from moving objects.

## 5.3 DEPTH ESTIMATION

Following point cloud prediction, we align predicted point clouds with ground-truth. These are then reprojected to each view, recording predicted and ground-truth depth values, with performance evaluated based on differences for valid points.

Table 3: Quantitative comparisons of depth estimation performance on RobustNeRF dataset.

| | Android | | Crab1 | | Crab2 | | Yoda | | Average | |
|---|---|---|---|---|---|---|---|---|---|---|
| | Abs Rel↓ | $\delta < 1.25$↑ | Abs Rel↓ | $\delta < 1.25$↑ | Abs Rel↓ | $\delta < 1.25$↑ | Abs Rel↓ | $\delta < 1.25$↑ | Abs Rel↓ | $\delta < 1.25$↑ |
| DUSt3R | 0.192 | 70.8 | 0.101 | 85.9 | 0.138 | 77.7 | 0.060 | 96.1 | 0.123 | 82.6 |
| MaSt3R | 0.491 | 61.0 | 0.182 | 66.2 | 0.372 | 54.8 | 0.150 | 73.7 | 0.299 | 63.9 |
| Fast3R | 0.213 | 64.7 | 0.204 | 57.1 | 0.255 | 57.2 | 0.110 | 83.5 | 0.196 | 65.6 |
| VGGT | 0.088 | 83.6 | 0.125 | 78.6 | 0.127 | 77.9 | 0.105 | 84.1 | 0.111 | 81.1 |
| VGTW(VGGT) | **0.049** | **96.9** | 0.091 | 88.7 | 0.049 | 96.0 | 0.057 | 96.3 | 0.062 | 94.5 |
| $\pi^3$ | 0.090 | 88.6 | 0.065 | 96.7 | 0.048 | 97.7 | 0.040 | 97.2 | 0.061 | 95.1 |
| VGTW($\pi^3$) | 0.063 | 91.2 | **0.030** | **100.0** | **0.046** | **97.8** | **0.024** | **100.0** | **0.041** | **97.3** |

Metrics include absolute relative error (Abs Rel ↓) and the percentage of predictions within a 1.25 threshold ($\delta < 1.25$ ↑).

Table 4 presents results on NeRF-on-the-go (unseen during training). DUSt3R achieves the best performance in low and high occlusion scenarios, yet our VGTW(VGGT) and VGTW($\pi^3$) variants deliver comparable performance while retaining the multi-frame efficiency of prior methods, with VGTW(VGGT) excelling in medium occlusion and VGTW($\pi^3$) leading in high occlusion, showing significant improvements over their original VGGT and $\pi^3$ counterparts.

Table 4: Quantitative comparisons of depth estimation performance on NeRF on-the-go dataset, averaged across scenes.

| | Low Occlusion | | Medium Occlusion | | High Occlusion | |
|---|---|---|---|---|---|---|
| | Abs Rel↓ | $\delta < 1.25$↑ | Abs Rel↓ | $\delta < 1.25$↑ | Abs Rel↓ | $\delta < 1.25$↑ |
| DUSt3R | **0.339** | **59.5** | 0.142 | 78.7 | **0.071** | **90.6** |
| MaSt3R | 0.941 | 37.2 | 0.240 | 63.9 | 0.163 | 74.2 |
| Fast3R | 0.524 | 47.8 | 0.155 | 76.4 | 0.119 | 81.2 |
| VGGT | 0.500 | 45.5 | 0.246 | 58.7 | 0.252 | 64.7 |
| VGTW(VGGT) | 0.346 | 55.9 | **0.125** | **82.2** | 0.220 | 71.1 |
| $\pi^3$ | 0.531 | 48.1 | 0.201 | 67.5 | 0.120 | 81.4 |
| VGTW($\pi^3$) | 0.405 | 52.7 | 0.152 | 74.9 | 0.076 | 88.1 |

On RobustNeRF, Table 3 highlights our methods' superiority, with VGTW($\pi^3$) achieving the lowest Abs Rel (e.g., 0.024 on Yoda) and perfect $\delta < 1.25$ scores in multiple scenes, reflecting enhanced accuracy over VGGT and $\pi^3$ due to effective distractor handling.

## 5.4 ABLATION STUDY

Table 5: Ablation study results tested on NeRF on-the-go dataset.

| $\mathcal{L}_{supp}$ | $\mathcal{L}_{cons}$ | $head_{mask}$ | Low Occlusion | | | | Medium Occlusion | | | | High Occlusion | | | | RobustNeRF Dataset | | | |
|---|---|---|---|---|---|---|---|---|---|---|---|---|---|---|---|---|---|---|
| | | | Acc↓ | Comp↓ | Overall↓ | NC↑ | Acc↓ | Comp↓ | Overall↓ | NC↑ | Acc↓ | Comp↓ | Overall↓ | NC↑ | Acc↓ | Comp↓ | Overall↓ | NC↑ |
| - | - | - | 0.039 | 0.057 | 0.048 | 0.663 | 0.051 | 0.120 | 0.085 | 0.631 | 0.032 | 0.262 | 0.147 | 0.624 | 0.021 | 0.045 | 0.033 | 0.684 |
| ✔ | - | - | 0.041 | 0.039 | 0.040 | 0.653 | 0.052 | 0.119 | 0.085 | 0.691 | 0.033 | **0.169** | **0.101** | 0.669 | 0.021 | 0.034 | 0.028 | 0.694 |
| ✔ | ✔ | - | 0.033 | 0.029 | 0.031 | 0.693 | **0.040** | 0.102 | 0.071 | 0.713 | **0.024** | 0.272 | 0.148 | 0.654 | 0.011 | 0.025 | 0.018 | 0.740 |
| ✔ | ✔ | ✔ | **0.033** | **0.029** | **0.031** | **0.695** | 0.040 | **0.095** | **0.068** | **0.722** | 0.025 | 0.228 | 0.126 | **0.693** | **0.011** | **0.025** | **0.018** | **0.740** |

We ablate components in point map estimation using the metrics: Acc↓, Comp↓, Overall↓ (average of Acc and Comp), and NC↑. We evaluate on NeRF-on-the-go (across occlusion levels) and RobustNeRF.

Table 5 shows that adding $\mathcal{L}$supp improves Comp (e.g., high occlusion: 0.262 to 0.169) and Overall (0.147 to 0.101). Incorporating $\mathcal{L}$cons further enhances Acc (e.g., low occlusion: 0.041 to 0.033) and NC (e.g., medium: 0.691 to 0.713). The full model with $head_{mask}$ achieves the best results across most metrics (e.g., medium Overall: 0.071 to 0.068), demonstrating synergy. On RobustNeRF, the mask head provides minor gains, as distractor regions do not constitute a large proportion and prior losses already reduce distractor confidence, enabling effective filtering at 50% points.

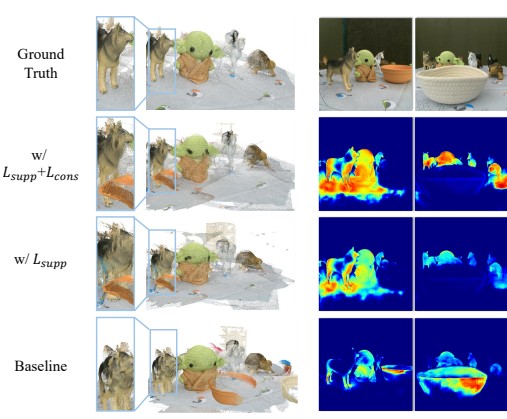

Figure 6: Ablation of using $\mathcal{L}$supp and $\mathcal{L}$cons.

Qualitatively, Fig. 6 shows the baseline assigns high confidence to distractors, preventing filtering and causing ghosting in static areas (e.g., wolf in bounding box). $\mathcal{L}_{supp}$ alone reduces distractor confidence but harms static features by lowering their confidence and alignment. Combining both losses minimizes distractor pollution while improving inter-frame consistency for better static alignment.

# 6 CONCLUSION

We present Visual Geometry Transformer in the Wild (VGTW), the first feed-forward method for robust 3D reconstruction from in-the-wild images with distractors. Our novel distractor-aware training strategy effectively separates static scene features from distractor-contaminated ones, ensuring geometric consistency. Additionally, our auxiliary mask prediction head accurately identifies and filters distractor regions, enabling clean, distractor-free point clouds in a single forward pass.

**Limitations.** We have not yet explored VGTW's performance on other 3D tasks, such as pose estimation or tracking. Furthermore, incorporating 3D label supervision could potentially enhance reconstruction accuracy.

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

# A  APPENDIX

## A.1  THE USE OF LARGE LANGUAGE MODELS (LLMs)

The research ideas and experimental design of this paper were conceived entirely by the authors without the use of LLMs. During manuscript preparation, we used GPT to assist with grammar checking and refinement of language for clarity and readability.

# B  EXTENDED EVALUATIONS FROM MAIN PAPER

## B.1  COMPARISON STUDIES

The detailed quantitative results in Tab.6–9 expand on the main text's aggregated metrics, offering per-scene breakdowns that reinforce VGTW's robustness in managing transient distractors across occlusion levels and datasets. These findings align with the Distractor-aware Training (DAT) strategy, which isolates contaminated features while preserving consistent scene elements.

On the NeRF On-the-go dataset, Tab.6 reports accuracy error (Acc $\downarrow$), completeness error (Comp $\downarrow$), and normal consistency (NC $\uparrow$). In high-occlusion Patio-High, VGTW(VGGT) achieves Acc of 0.0332 (vs. VGGT's 0.0365) and NC of 0.6308 (vs. 0.5862), prioritizing accurate static geometry despite minor Comp increases. VGTW($\pi^3$) reduces Acc from 0.0471 to 0.0200 and boosts NC from 0.7340 to 0.7762. Averaged improvements reach 20-40% in error metrics for high-occlusion cases, confirming generalization without optimization. Tab.7 evaluates depth estimation via absolute relative error (Abs Rel $\downarrow$) and $\delta < 1.25$ ($\uparrow$). In medium-occlusion Patio, VGTW(VGGT) lowers Abs Rel from 0.275 to 0.149 and raises $\delta < 1.25$ from 50.9% to 77.4%, highlighting the cross-view consistency loss's role in stable predictions.

For RobustNeRF, Tab.8 and Tab.9 show significant gains with pixel-level annotations. In Tab.8, VGTW(VGGT) halves Acc for Android (0.0269 to 0.0069) and increases NC to 0.7905 (vs. 0.7462). Tab.9 boosts $\delta < 1.25$ to over 96% in several scenes, underscoring DAT's feature separation for superior fidelity.

To assess camera pose estimation robustness, we evaluate VGTW on the NeRF On-the-go datase Ren et al. (2024), featuring scenes with varying occlusion from transient distractors. Tab.A.1 presents results via Relative Pose Error for translation ($RPE_{trans}$) and rotation ($RPE_{rot}$), plus Absolute Trajectory Error (ATE)—all lower-is-better metrics. Against baseline VGGT, VGTW shows consistent gains, especially in $RPE_{rot}$ and ATE, with 0.5-1.5% average reductions in low-occlusion scenes (e.g., Mountain, Fountain) and 1-2% in medium/high-occlusion ones (e.g., Patio-High, ATE from 0.5885 to 0.5741). This arises from VGTW's distractor-aware attention, reducing transient feature pollution for better alignments under challenges. Minor $RPE_{trans}$ trade-offs appear in select scenes (e.g., Fountain, Spot), but overall superiority highlights strong generalization to inconsistent real-world views without efficiency loss.

Fig.7 extends main-text visualizations, displaying cleaner point maps on RobustNeRF and NeRF On-the-go scenes. VGTW removes distractor artifacts persistent in baselines, yielding coherent static geometry aligned with ground truth, supporting claims of enhanced robustness in in-the-wild scenarios.

Table 6: Quantitative comparisons of 3D reconstruction performance on NeRF on-the-go dataset.

| | Low Occlusion | | | | | | Medium Occlusion | | | | | | High Occlusion | | | | | |
| | Mountain | | | Fountain | | | Corner | | | Patio | | | Spot | | | Patio-High | | |
| | Acc↓ | Comp↓ | NC↑ | Acc↓ | Comp↓ | NC↑ | Acc↓ | Comp↓ | NC↑ | Acc↓ | Comp↓ | NC↑ | Acc↓ | Comp↓ | NC↑ | Acc↓ | Comp↓ | NC↑ |
|---|---|---|---|---|---|---|---|---|---|---|---|---|---|---|---|---|---|---|
| DUSt3R | 0.0150 | **0.0120** | 0.6896 | 0.0697 | 0.0589 | 0.6735 | 0.0458 | 0.0504 | **0.8195** | 0.0422 | 0.1539 | 0.7062 | 0.0171 | 0.0605 | **0.8312** | 0.0343 | 0.1428 | 0.7644 |
| MaSt3R | 0.0190 | 0.0570 | 0.6447 | 0.0694 | 0.1835 | 0.6273 | 0.0580 | 0.0872 | 0.7535 | 0.0694 | 0.1835 | 0.6273 | 0.0248 | 0.1835 | 0.7775 | 0.0298 | 0.2476 | 0.7236 |
| Fast3R | 0.0168 | 0.0187 | 0.6858 | 0.0565 | 0.0786 | 0.5921 | 0.0564 | 0.0451 | 0.7706 | 0.0565 | **0.0786** | 0.5921 | 0.0162 | 0.0725 | 0.7701 | 0.0430 | **0.1177** | 0.6685 |
| VGGT | 0.0187 | 0.0239 | 0.6960 | 0.0600 | 0.0901 | 0.6299 | 0.0585 | 0.0759 | 0.6303 | 0.0425 | 0.1635 | 0.6325 | 0.0271 | 0.1391 | 0.6626 | 0.0365 | 0.3841 | 0.5862 |
| VGTW(VGGT) | 0.0082 | 0.0151 | **0.7375** | 0.0573 | **0.0423** | 0.6523 | 0.0397 | 0.0484 | 0.7403 | **0.0408** | 0.1414 | **0.7070** | **0.0160** | 0.0502 | 0.7545 | 0.0332 | 0.4059 | 0.6308 |
| $\pi^3$ | 0.0124 | 0.0234 | 0.7065 | 0.0545 | 0.0783 | 0.6906 | 0.0389 | 0.0560 | 0.7624 | 0.1280 | 0.1348 | 0.5888 | 0.0236 | 0.0374 | 0.7716 | 0.0471 | 0.1181 | 0.7340 |
| VGTW($\pi^3$) | **0.0054** | 0.0227 | 0.5511 | **0.0522** | 0.0425 | **0.6994** | **0.0186** | **0.0319** | 0.7620 | 0.0499 | 0.1142 | 0.6681 | 0.0176 | **0.0317** | 0.6929 | **0.0200** | 0.1192 | **0.7762** |

Table 7: Quantitative comparisons of depth estimation performance on NeRF on-the-go dataset.

| | Low Occlusion | | | | Medium Occlusion | | | | High Occlusion | | | |
| | Mountain | | Fountain | | Corner | | Patio | | Spot | | Patio-High | |
| | Abs Rel↓ | $\delta < 1.25$↑ | Abs Rel↓ | $\delta < 1.25$↑ | Abs Rel↓ | $\delta < 1.25$↑ | Abs Rel↓ | $\delta < 1.25$↑ | Abs Rel↓ | $\delta < 1.25$↑ | Abs Rel↓ | $\delta < 1.25$↑ |
|---|---|---|---|---|---|---|---|---|---|---|---|---|
| DUSt3R | 0.163 | 75.0 | 0.514 | 43.9 | 0.116 | 84.0 | 0.168 | 73.4 | 0.020 | 99.8 | 0.122 | 81.3 |
| MaSt3R | 0.671 | 32.5 | 1.211 | 41.8 | 0.179 | 77.3 | 0.301 | 50.5 | 0.070 | 92.5 | 0.256 | 55.8 |
| Fast3R | 0.367 | 58.5 | 0.681 | 37.1 | 0.068 | 94.4 | 0.242 | 58.3 | 0.022 | 99.7 | 0.216 | 62.7 |
| VGGT | 0.331 | 49.8 | 0.668 | 41.1 | 0.216 | 66.4 | 0.275 | 50.9 | 0.081 | 89.8 | 0.422 | 39.5 |
| VGTW(VGGT) | 0.196 | 68.5 | 0.496 | 43.3 | 0.100 | 86.9 | 0.149 | 77.4 | 0.025 | 100.0 | 0.415 | 42.1 |
| $\pi^3$ | 0.306 | 53.3 | 0.756 | 42.9 | 0.189 | 71.9 | 0.213 | 63.1 | 0.026 | 100.0 | 0.213 | 62.7 |
| VGTW($\pi^3$) | 0.247 | 62.7 | 0.562 | 42.7 | 0.120 | 81.47 | 0.183 | 68.4 | 0.016 | 100.0 | 0.136 | 76.2 |

Table 8: Quantitative comparisons of 3D reconstruction performance on RobustNeRF dataset.

| | Android | | | Crab1 | | | Crab2 | | | Yoda | | |
| | Acc↓ | Comp↓ | NC↑ | Acc↓ | Comp↓ | NC↑ | Acc↓ | Comp↓ | NC↑ | Acc↓ | Comp↓ | NC↑ |
|---|---|---|---|---|---|---|---|---|---|---|---|---|
| DUSt3R | 0.0541 | 0.1412 | 0.6986 | 0.0228 | 0.0405 | 0.6534 | 0.0281 | 0.0544 | 0.6592 | 0.0183 | 0.0354 | 0.7737 |
| MaSt3R | 0.0518 | 0.2940 | 0.6961 | 0.0261 | 0.4031 | 0.5649 | 0.0370 | 0.3265 | 0.5783 | 0.0369 | 0.5264 | 0.6226 |
| Fast3R | 0.0285 | 0.1335 | 0.6930 | 0.0406 | 0.1167 | 0.5963 | 0.0386 | 0.1055 | 0.6239 | 0.0273 | 0.0619 | 0.7280 |
| VGGT | 0.0269 | 0.0331 | 0.7462 | 0.0196 | 0.0462 | 0.6332 | 0.0174 | 0.0459 | 0.6618 | 0.0183 | 0.0550 | 0.6944 |
| VGTW(VGGT) | 0.0069 | 0.0174 | **0.7905** | 0.0169 | 0.0345 | 0.6663 | **0.0094** | 0.0198 | 0.7392 | 0.0121 | 0.0285 | 0.7639 |
| $\pi^3$ | 0.0154 | 0.0175 | 0.7500 | 0.0214 | 0.0188 | **0.7232** | 0.0151 | 0.0124 | **0.7428** | 0.0149 | 0.0155 | **0.7988** |
| VGTW($\pi^3$) | **0.0045** | **0.0067** | 0.7802 | **0.0099** | **0.0108** | 0.6854 | **0.0127** | 0.0111 | 0.6880 | **0.0105** | **0.0102** | 0.7175 |

Table 9: Quantitative comparisons of depth estimation performance on RobustNeRF dataset.

| | Android | | Crab1 | | Crab2 | | Yoda | |
| | Abs Rel↓ | $\delta < 1.25$↑ | Abs Rel↓ | $\delta < 1.25$↑ | Abs Rel↓ | $\delta < 1.25$↑ | Abs Rel↓ | $\delta < 1.25$↑ |
|---|---|---|---|---|---|---|---|---|
| DUSt3R | 0.192 | 70.8 | 0.101 | 85.9 | 0.138 | 77.7 | 0.060 | 96.1 |
| MaSt3R | 0.491 | 61.0 | 0.182 | 66.2 | 0.372 | 54.8 | 0.150 | 73.7 |
| Fast3R | 0.213 | 64.7 | 0.204 | 57.1 | 0.255 | 57.2 | 0.110 | 83.5 |
| VGGT | 0.088 | 83.6 | 0.125 | 78.6 | 0.127 | 77.9 | 0.105 | 84.1 |
| VGTW(VGGT) | **0.049** | **96.9** | 0.091 | 88.7 | 0.049 | 96.3 | 0.057 | 96.3 |
| $\pi^3$ | 0.090 | 88.6 | 0.065 | 96.7 | 0.048 | 97.7 | 0.040 | 97.2 |
| VGTW($\pi^3$) | 0.063 | 91.2 | **0.030** | **100.0** | **0.046** | **97.8** | **0.024** | **100.0** |

Table 10: Quantitative comparisons of camera pose estimation performance on NeRF on-the-go dataset.

| | Low Occlusion | | | | | | Medium Occlusion | | | | | | High Occlusion | | | | | |
| | Mountain | | | Fountain | | | Corner | | | Patio | | | Spot | | | Patio-High | | |
| | $RPE_{trans}$↓ | $RPE_{rot}$↓ | ATE↓ | $RPE_{trans}$↓ | $RPE_{rot}$↓ | ATE↓ | $RPE_{trans}$↓ | $RPE_{rot}$↓ | ATE↓ | $RPE_{trans}$↓ | $RPE_{rot}$↓ | ATE↓ | $RPE_{trans}$↓ | $RPE_{rot}$↓ | ATE↓ | $RPE_{trans}$↓ | $RPE_{rot}$↓ | ATE↓ |
|---|---|---|---|---|---|---|---|---|---|---|---|---|---|---|---|---|---|---|
| VGGT | 6.9727 | 89.2598 | 1.5571 | 11.4844 | 122.2366 | 1.7529 | 2.1168 | 49.6358 | 1.1096 | 2.7503 | 58.5429 | 1.2225 | 5.9165 | 51.9370 | 1.0209 | 4.2482 | 68.4159 | 0.5885 |
| VGTW | 6.9688 | 88.6475 | 1.5642 | 11.5001 | 122.0872 | 1.7380 | 2.1135 | 49.4836 | 1.1036 | 2.7475 | 58.3305 | 1.2116 | 5.9331 | 51.8181 | 1.0129 | 4.2464 | 68.2577 | 0.5741 |

## B.2 ABLATION STUDIES

Tab.11–12 provide per-scene ablations of VGTW components ($L_{supp}$, $L_{cons}$, mask head), building on main-text summaries to confirm DAT's synergistic effects without 3D supervision. On NeRF On-the-go (Tab.11), adding $L_{supp}$ improves Comp (e.g., Patio-High: 0.3841 to 0.2298) and NC (0.5862 to 0.6165). Combining $L_{cons}$ refines further (NC to 0.5949), while the full model with mask head achieves optimal NC (0.6308), yielding 10-30% gains across levels.

On RobustNeRF (Tab.12), $L_{supp}$ reduces Android Acc to 0.0204 and boosts NC to 0.7523. With $L_{cons}$, Acc drops to 0.0069 and NC to 0.7903; the mask head finalizes averaged NC at 0.7400 (vs. baseline 0.6839), with consistent improvements like Yoda NC from 0.6944 to 0.7639. These affirm DAT's efficiency and the RobustNeRF-Mask dataset's value for 2D-supervised distractor-aware attention.

Table 11: Ablation study results on NeRF on-the-go.

| | | | Low Occlusion | | | | | | Medium Occlusion | | | | | | High Occlusion | | | | | |
| | | | Mountain | | | Fountain | | | Corner | | | Patio | | | Spot | | | Patio-High | | |
| $L_{supp}$ | $L_{cons}$ | $head_{mask}$ | Acc↓ | Comp↓ | NC↑ | Acc↓ | Comp↓ | NC↑ | Acc↓ | Comp↓ | NC↑ | Acc↓ | Comp↓ | NC↑ | Acc↓ | Comp↓ | NC↑ | Acc↓ | Comp↓ | NC↑ |
|---|---|---|---|---|---|---|---|---|---|---|---|---|---|---|---|---|---|---|---|---|
| - | - | - | 0.0187 | 0.0239 | 0.6960 | 0.0600 | 0.0901 | 0.6299 | 0.0585 | 0.0759 | 0.6303 | 0.0425 | 0.1635 | 0.6325 | 0.0271 | 0.1391 | 0.6626 | 0.0365 | 0.3841 | 0.5862 |
| ✓ | - | - | 0.0195 | 0.0216 | 0.6980 | 0.0620 | 0.0567 | 0.6077 | 0.0530 | 0.0624 | 0.6915 | 0.0507 | 0.1754 | 0.6899 | 0.0268 | 0.1090 | 0.7218 | 0.0398 | 0.2298 | 0.6165 |
| ✓ | ✓ | - | 0.0082 | 0.0151 | 0.7373 | 0.0573 | 0.0427 | 0.6481 | 0.0394 | 0.0504 | 0.7388 | 0.0400 | 0.1544 | 0.6878 | 0.0149 | 0.1142 | 0.7127 | 0.0324 | 0.4306 | 0.5949 |
| ✓ | ✓ | ✓ | 0.0082 | 0.0151 | 0.7375 | 0.0573 | 0.0423 | 0.6523 | 0.0397 | 0.0484 | 0.7403 | 0.0408 | 0.1414 | 0.7040 | 0.0160 | 0.0502 | 0.7545 | 0.0332 | 0.4059 | 0.6308 |

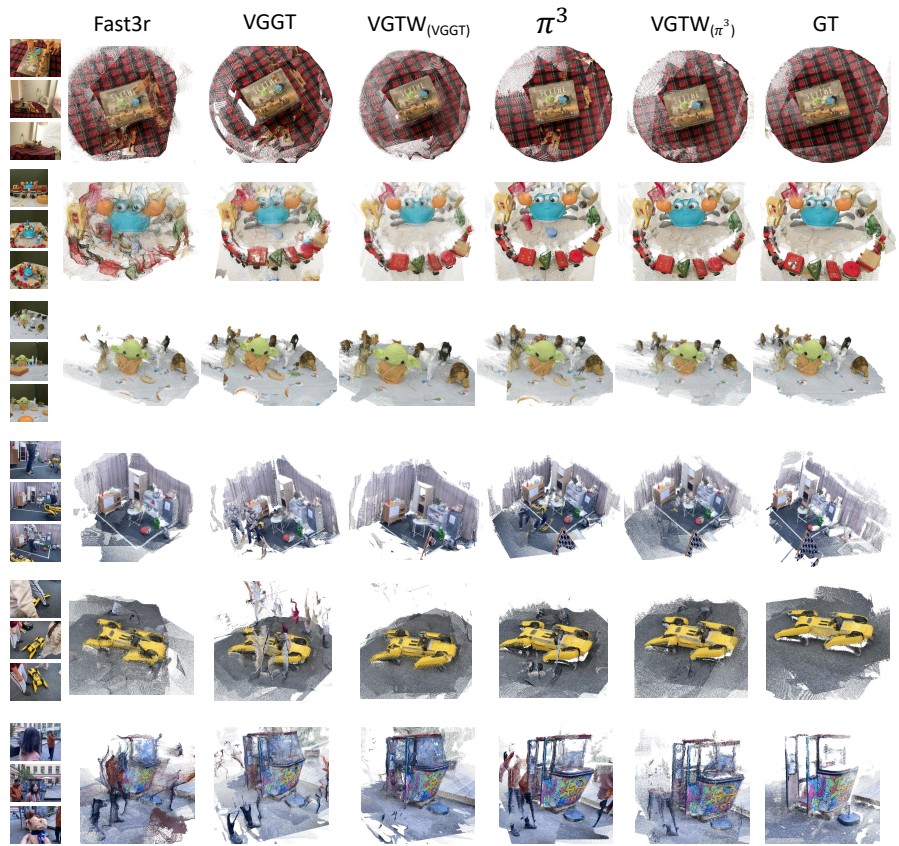

Figure 7: Qualitative results of point map estimation. Rows 1-3: RobustNeRF dataset (android, crab2, yoda). Rows 4-6: NeRF On-the-go (corner, spot, patio-high).

Table 12: Ablation study results tested on RobustNeRF dataset.

| $\mathcal{L}_{supp}$ | $\mathcal{L}_{cons}$ | $head_{mask}$ | Android | | | Crab1 | | | Crab2 | | | Yoda | | | Averaged | | |
|---|---|---|---|---|---|---|---|---|---|---|---|---|---|---|---|---|---|
| | | | Acc↓ | Comp↓ | NC↑ | Acc↓ | Comp↓ | NC↑ | Acc↓ | Comp↓ | NC↑ | Acc↓ | Comp↓ | NC↑ | Acc↓ | Comp↓ | NC↑ |
| - | - | - | 0.0269 | 0.0331 | 0.7462 | 0.0196 | 0.0462 | 0.6332 | 0.0174 | 0.0459 | 0.6618 | 0.0183 | 0.0550 | 0.6944 | 0.0206 | 0.0451 | 0.6839 |
| ✔ | - | - | 0.0204 | 0.0301 | 0.7523 | 0.0224 | 0.0326 | 0.6513 | 0.0201 | 0.0329 | 0.6649 | 0.0203 | 0.0413 | 0.7071 | 0.0208 | 0.0342 | 0.6939 |
| ✔ | ✔ | - | 0.0069 | 0.0174 | 0.7903 | 0.0169 | 0.0346 | 0.6662 | 0.0094 | 0.0198 | 0.7391 | 0.0121 | 0.0285 | 0.7638 | 0.0113 | 0.0251 | 0.7399 |
| ✔ | ✔ | ✔ | 0.0069 | 0.0174 | **0.7905** | 0.0169 | 0.0345 | 0.6663 | **0.0094** | 0.0198 | 0.7392 | 0.0121 | 0.0285 | 0.7639 | 0.0113 | 0.0251 | 0.7400 |

### B.3 CROSS-DOMAIN EVALUATION

The cross-domain evaluations in Fig.8 and Fig.9 demonstrate VGTW's versatility across dynamic and static scenarios, extending its applicability beyond the primary benchmarks and reinforcing its generalization without additional training.

**Dynamic video.** In dynamic videos from the DAVIS dataset Pont-Tuset et al. (2017), Fi.8 illustrates VGTW's effectiveness in filtering transient motion. For instance, in sequences featuring moving camels, VGTW selectively removes the walking camel while preserving the stationary one, resulting in a clean reconstruction of the static background. In contrast, VGGT produces "ghosting" artifacts from the motion, blending inconsistent features into spurious geometry. Similarly, in videos with vehicles, VGTW suppresses dynamic elements like motor bike rushing figures, maintaining coherent static structures such as buildings or landscapes, thus highlighting the DAT strategy's robustness in handling temporal inconsistencies.

**Static scene.** For static scenes, evaluations on the DTU dataset Jensen et al. (2014) and the static subset of NeRF On-the-go Ren et al. (2024) reveal that VGTW achieves reconstruction quality comparable to VGGT, even in distractor-free inputs. Fig.9 shows that VGTW preserves fine details effectively, with enhanced performance in certain cases, such as sharper texture reconstruction on building facades or objects like sculptures, where it reduces minor artifacts and improves surface fidelity. This parity confirms that VGTW's distractor-aware mechanisms do not compromise performance in ideal conditions, ensuring broad utility across domains.

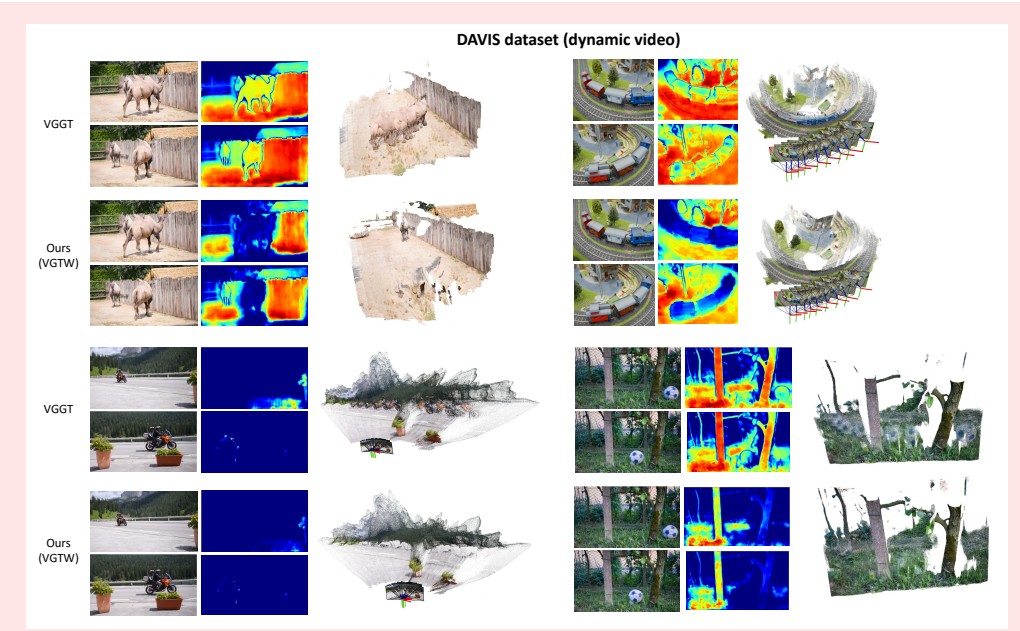

Figure 8: Qualitative results of VGTW and VGGT on DAVIS Pont-Tuset et al. (2017) dataset.

### B.4 SENSITIVITY TO POOR ANNOTATION

To evaluate VGTW's sensitivity to label noise, we constructed a "dirty" dataset by mislabeling 10% of distractor masks in RobustNeRF-Mask as static regions, then retrained under the same settings. We assessed mask estimation on the annotated Corner subset of NeRF On-the-go, where paired distractor images and masks were added. Tab.A.1 shows a decline in accuracy for the poorly trained VGTW, with F1 dropping from 0.5896 to 0.4003, primarily due to reduced recall (0.5738 to 0.3375), indicating challenges in detecting transients amid noisy labels. Nevertheless, Fig.10 illustrates that this variant still outperforms baseline VGGT in suppressing distractor confidence, producing cleaner point clouds in scenes like Corner, Spot, and Patio. These results highlight VGTW's resilience to imperfect annotations, preserving effective distractor isolation even with contaminated training data.

Table 13: Distractor mask estimation performance comparison on the Corner Subset of the NeRF-on-the-go Dataset Ren et al. (2024) (Higher is Better ↑)

| Model | Acc↑ | Precision↑ | Recall↑ | F1↑ | IoU↑ | Dice↑ |
|---|---|---|---|---|---|---|
| VGTW (poor trained) | 0.9505 | 0.6013 | 0.3375 | 0.4003 | 0.2971 | 0.4003 |
| VGTW | 0.9559 | 0.6304 | 0.5738 | 0.5896 | 0.4748 | 0.5896 |

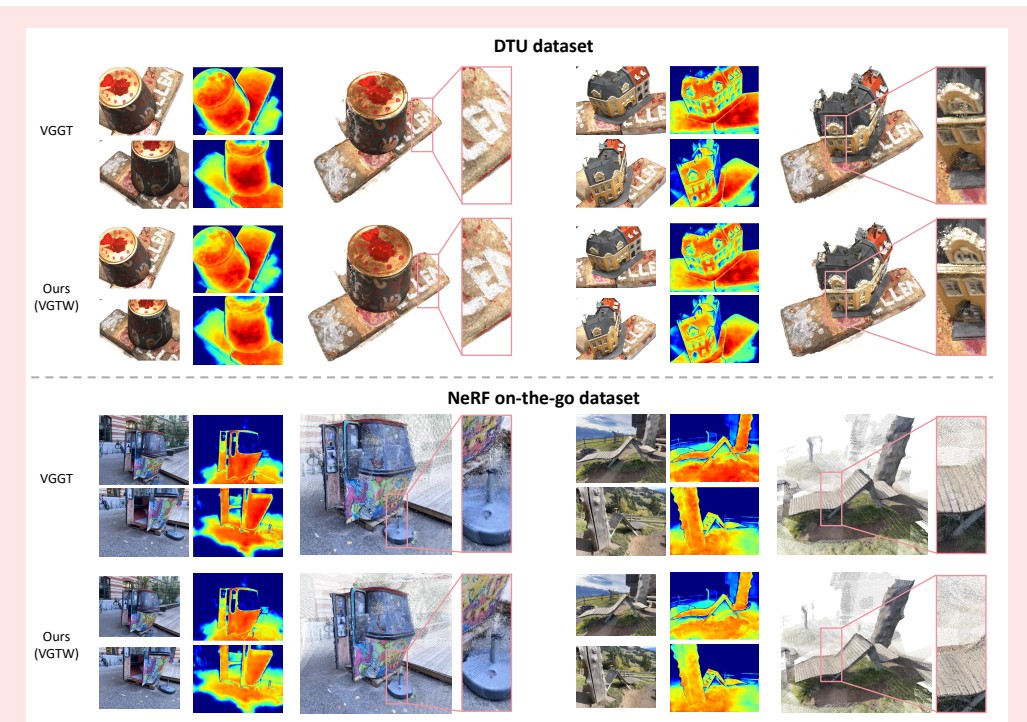

Figure 9: Qualitative results of point map estimation on static scenarios. Rows 1-2: NeRF On-the-go dataset Ren et al. (2024). Rows 3-4: DTU dataset Jensen et al. (2014).

## C    COMPARISON WITH DYNAMIC VIDEO RECONSTRUCTION PIPELINE.

While MonST3R Zhang et al. (2024b) demonstrates effectiveness on dynamic video sequences in controlled settings, it struggles on the NeRF On-the-go dataset Ren et al. (2024), which features complex distractors—including both dynamic and transient objects—within unordered image collections. As illustrated in Fig. 11, MonST3R fails to adequately filter out distractors, resulting in persistent artifacts, while also yielding poor static region reconstructions marked by severe misalignments. This limitation arises from its DUSt3R-style pairwise processing, which, as shown in Tab. 14, causes runtime to scale significantly with increasing input images (e.g., from 10 to 50 views). Similarly, Easi3R Chen et al. (2025) can effectively detect distractors via aggregated cross-attention maps in scenarios with small viewpoint shifts; however, it is prone to misclassifying static regions as distractors in cases with large parallax, leading to over-filtering and incomplete reconstructions, as exemplified in the patio scene (Fig. 11). This stems from its reliance on limited pairwise correlations and lack of global multi-view perception. In contrast, VGTW maintains efficiency comparable to VGGT while leveraging distractor-aware attention and cross-view consistency to robustly suppress transients without erroneously discarding static geometry, enabling feed-forward reconstruction superior in both fidelity and scalability across diverse in-the-wild settings.

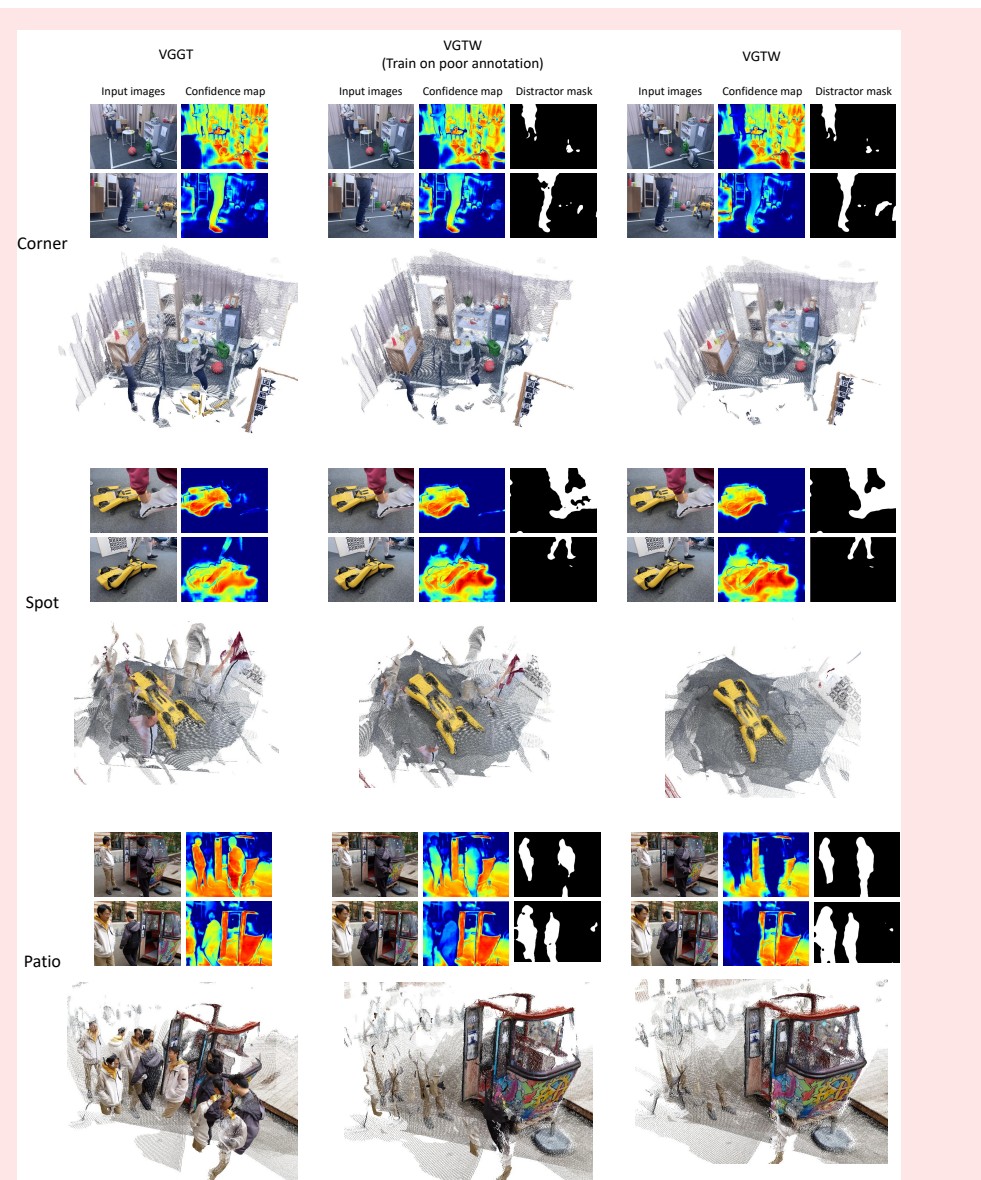

Figure 10: Qualitative results of VGTW and its variant trained on dirty annotated RobustNeRF-mask dataset.

## D DATA PREPARATION DETAILS.

**Annotation.** Our dataset annotations are entirely human-driven, as automatic labeling methods like optical-flow-based techniques or SAM2 propagation are not suitable for annotating RobustNeRF. This is because objects in the dataset do not consistently move across frames. The dataset includes transient objects, where some objects appear in one frame but are absent in others. Therefore, we rely heavily on manual inspection to identify inconsistencies between frames and determine which regions are distractors. SAM2 is then used to generate masks for these identified distractor objects. **Choosing 50% as filter threshold.** As shown in Fig.12, Choosing keep ratio of 50% is an appropriate threshold in getting point cloud. It balances preserving the completeness of the target scene while effectively filtering out most distractors and background artifacts.

Table 14: Inference time of different models under various frame numbers.

| Model | Frame number | | |
|---|---|---|---|
| | 5 | 10 | 20 |
| MonST3R | 14.82s | 28.94s | 70.99s |
| VGGT | 0.36s | 0.65s | 1.38s |
| VGTW | 0.38s | 0.70s | 1.45s |

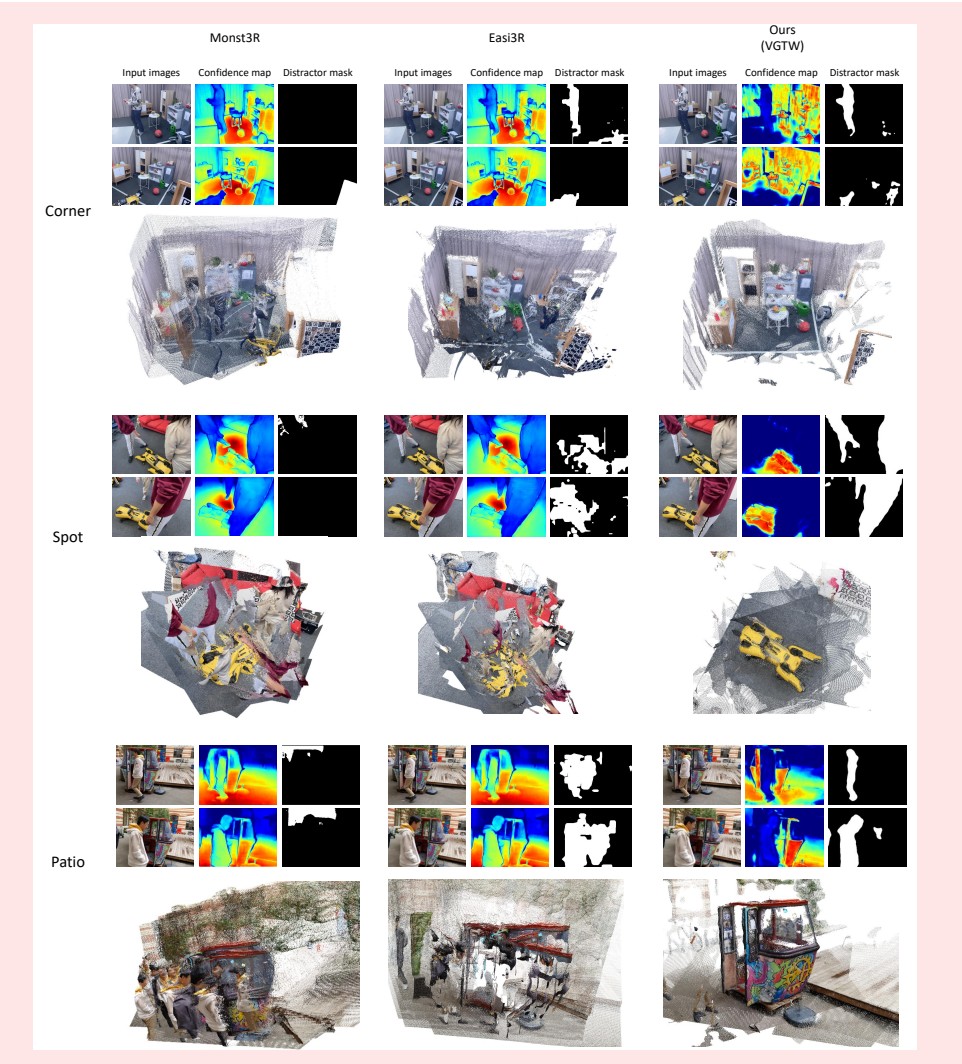

Figure 11: Qualitative results of VGTW and MonST3R Zhang et al. (2024b) on NeRF on-the-go.

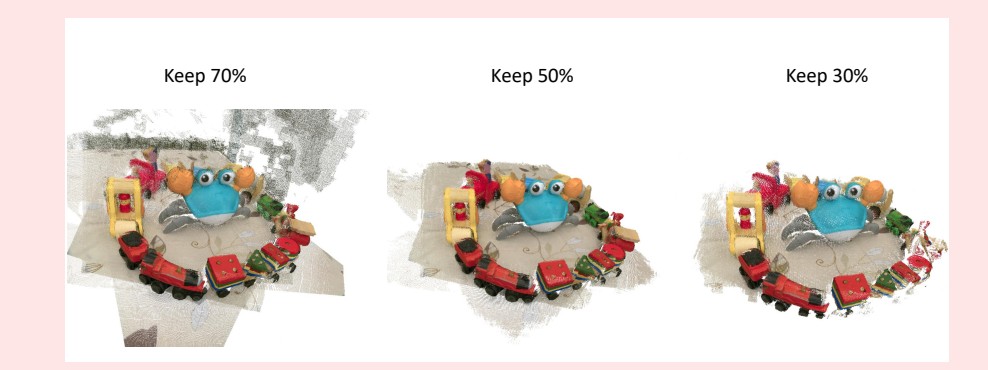

Figure 12: Different keeping ratio when generating GT point cloud, raning from 30% to 70%.

