# OpenReview forum: "Visual Geometry Transformer in the Wild: Distractor-Free 3D Reconstruction"
_ICLR.cc/2026/Conference — Submitted to ICLR 2026_

### Official Review · Reviewer_qMPh · 2025-10-31

**Soundness:** 2
**Presentation:** 2
**Contribution:** 2
**Rating:** 2
**Confidence:** 4

**Summary:**

This paper presents *VGTW (Visual Geometry Transformer in the Wild)*, a feed-forward transformer-based pipeline that aims to produce distractor-free 3D reconstructions from multi-view images. The authors fine-tune existing feed-forward 3D models (VGGT, π3) with LoRA and introduce two losses (Distractor Suppression and Cross-View Consistency) plus a distractor mask prediction head. They claim improved robustness to transient distractors and strong results on NeRF-on-the-go and RobustNeRF.

While the paper has some strengths, such as insightful analysis of attention leaking to distractors, intuitive loss formulations, there are several concerns. The claims about prior feed-forward methods appear overstated, discussion about several highly related work is missing, and the evaluation is limited and sometimes unclear.

Overall, I would recommend reject as the contribution is incremental, the evaluation has gaps, and some claims are overstated.

**Strengths:**

1. **Insightful analysis of attention and loss design.** The observation that attention leaks to distractors is both insightful and clearly illustrated in Figure. The design of the two losses (Distractor Suppression and Cross-View Consistency) demonstrates benefits for achieving distractor-free predictions.

2. **Clarity and presentation.** The paper is generally well-written, with clear figures and intuitive formulations for the proposed losses, and the methodology is easy to follow.

3. **Strong experimental results.** The method achieves improved metrics on the RobustNeRF and NeRF-on-the-go datasets, showing its effectiveness in handling scenes with distractors.

**Weaknesses:**

1. **Overstated claim.** The paper repeatedly suggests that VGGT/π3 conceptually cannot handle dynamic scenes. This is not true, as existing feed-forward architectures are already designed to be robust to non-static inputs (especially π3). Therefore, the premise of the paper appears overstated.

2. **Missing related work.** The paper overlooks directly relevant studies that adapt DUSt3R-style frameworks for dynamic scenes, such as Monst3R [a] and Easi3R [b]. In particular, Easi3R discusses attention mechanisms for handling dynamic objects in reconstruction. This omission makes the contribution appear less novel.

3. **Limited and questionable evaluation.**
- Dataset mismatch: Both datasets were originally designed for "distractor-free" NeRF-based **novel view synthesis** rather than feed-forward reconstruction evaluation. The paper mentions that ground truth is "generated using pretrained π³ on distractor-free images" but does not clarify:
  a. During inference, how are "distractor-free" point clouds obtained, are they using confidence filtering?
  b. If so, what confidence threshold is used (0.5 in L358 in training section?), and is there a specific reason for choosing that threshold?
  c. The methods still output distractor point clouds without explicit occlusion area generation. Would simply removing the distractor regions  and comparing it with "GT" point cloulds be a fair evaluation setting?
- Small scale: Only a few cases/scenes are used for evaluation, which seems insufficient for feed-forward methods that claim generalization.

---

[a] Zhang, J., Herrmann, C., Hur, J., Jampani, V., Darrell, T., Cole, F., Sun, D., & Yang, M.-H. (2025). *MonST3R: A Simple Approach for Estimating Geometry in the Presence of Motion*. In ICLR 2025.
[b] Chen, X., Chen, Y., Xiu, Y., Geiger, A., & Chen, A. (2025). *Easi3R: Estimating Disentangled Motion from DUSt3R Without Training*. In ICCV 2025.

**Questions:**

1. The idea of generating distractor mask is interesting. Could the authors provide qualitative results for the mask? Additionally, I am interested in seeing how using only the mask head (without distractor-aware training, or just use the mask for pretrained π3 predictions) would affect the results.

2. Regarding training, could the authors specify the computational resources used (e.g., number of GPUs, total training time, and learning rate)? It also appears that the fine-tuning is performed without 3D supervision. Could the authors clarify how much this affects the 3D reconstruction performance?

3. Could the authors explain the evaluation procedure in more detail, as mentioned in *Weakness 3*?

4. For *Tables 1–3*, the authors might consider adding average scores to facilitate easier comparison.

---

> ### Author Response · Authors · 2025-11-24
> **Response to Reviewer qMPh (Part I)**
>
> We thank the reviewer for your detailed and constructive feedback on our submission. We appreciate the reviewer pointing out the omissions in our related works organization; in the responses below, we clarify the purpose of this study and its positioning within the related literature. If the reviewer has any aspects that remain unclear, we are more than happy to provide further explanations.
>
> ### >> **Q1 Claims about VGGT/$\pi^3$**
> $ \color{red} {A1:} $ Thanks! We agree that VGGT and $\pi^3$ can handle dynamic scenes. However, our key argument is that **they cannot perform distractor-free reconstructions**, which is the specific task we target.
> **Task Difference: Distractor-free reconstructions vs. Dynamic scenes**
> In fact, Dynamic scenes and distractor-free reconstructions are fundamentally different tasks:
> - **Dynamic scenes**: Within a continuous stream of frames, part of the scene is moving. The goal is to reconstruct both the static and dynamic components.
> - **Distractor-free reconstructions**: Objects do not need to move consistently across frames. Some could appear in one view but not in another (e.g., transient or occluded objects in datasets such as RobustNeRF and NeRF-on-the-go). The objective is to reconstruct only the static scene, filtering out distractors.
>
> **Why VGGT/$\pi^3$ Cannot Perform Distractor-free Reconstructions**
> VGGT/$\pi^3$ can do dynamic reconstriuction to some degree, but not our task.
> VGGT/$\pi^3$ are trained on both static and dynamic scene datasets. VGGT includes a tracking head to manage motion. But neither VGGT nor $\pi^3$is trained on the type of data required for distractor-free reconstruction.
> *Our experiments confirm this*: as shown in both qualitative and quantitative comparisons in `Tables 6–10 and Figures 7–8`, VGGT and $\pi^3$ fail to produce clean, distractor-free point clouds.
>
> ### >> **Q2 Missing related work**
> $ \color{red} {A2:} $ Thank you for the reminder. We have added citations to MonST3R and Easi3R in `Sec. 2.2`. These works are thought-provoking, exploring 4D reconstruction from dynamic videos. However, it is important to clarify the differences between their approaches and ours:
>
> - **Task Difference**: : Dynamic reconstruction from videos is fundamentally different from our task. We focus on transient distractors-objects that appear in some views but are absent in others. As noted in `A1`, these are not dynamic objects but **inconsistent elements across viewpoints**. An example is shown in `Fig. 11`.
> - **Paradigm Difference**: DUSt3R-style networks perform **pairwise reconstruction**. However, VGTW **processes multiple images jointly** in a single feed-forward pass.
> - **Distinct Approaches to Attention Handling**: VGTW implicitly fine-tunes attention via DAT to isolate distractors (`Sec. 4.1`), while Easi3R explicitly analyzes and modifies pre-trained attention maps at inference to suppress dynamic regions.
>
> ### >> **Q3 Dataset annotation & Evaluation details**
> $ \color{red}{A3:} $
> Thank you for your questions. Here are clarifications regarding your concerns:
>
> **Dataset choice**:
> Thanks for raising this concern. We believe our dataset selection is appropriate for the task. Our work focuses on feed-forward, in-the-wild 3D reconstruction; there is **no established benchmark exists**. RobustNeRF and NeRF-on-the-go are the *closest available tasks* there for in-the-wild novel view synthesis. These datasets include diverse distractor types (both dynamic and transient objects) and provide paired distractor-free images. Therefore, we adopt these datasets and generate the required ground truth for our purpose.
>
> **a. Generation of "distractor-free" point clouds**:
> We use $\pi_3$ model to produce clean images without distractors. From these images, we build point clouds in two steps. First, we apply a confidence threshold of 0.5 to filter uncertain points. Then, we remove remaining distractors using the predicted mask for precise refinement.
>
> **b. Confidence Threshold**:
> The 0.5 threshold is chosen because it is a common value in feed-forward 3D reconstruction methods (e.g., VGGT), effectively discarding low-confidence predictions while retaining useful information.
>
> **c. Comparison after removing distractor**:
> Simply removing distractor regions helps to some extent, but it is not as effective as our method. To demonstrate this, we **use ground-truth distractor masks and filter distractor regions after reconstruction**. We conduct additional experiments on an unseen dataset by annotating the Corner subset of NeRF-on-the-go and applying this “simple removal” strategy. The results show:
> - VGGT + GT mask performs better than plain VGGT.
> - VGTW further improves reconstruction quality compared to VGGT + GT that “simply removing distractor regions”.
> | | Acc↓ | Comp↓ | NC↑ |
> |-|-|-|-|
> | VGGT|0.0585|0.0759|0.6303|
> | VGGT+GT mask (**simple removal**)|0.0459|0.0598|0.7280|
> | VGTW | 0.0397 |0.0484|0.7403|
> | VGTW+GT mask| 0.0314| 0.0448 | 0.7573 |

---

> > ### Author Response · Authors · 2025-11-24
> > **Response to Reviewer qMPh (Part Ⅱ)**
> >
> > ### >> **Q4 Expanded scale of evaluation**
> > $ \color{red} {A4:} $ Thank you for your interest in our model's generalizability. We have added generalization tests across various datasets. Specifically, DAVIS [1] demonstrates our method's effectiveness on dynamic videos, as shown in `Fig.8`, where VGTW successfully removes dynamic objects from the cross-view reconstructed point cloud. Additionally, tests on the DTU [2] and NeRF-on-the-go static subsets [3] confirm strong reconstruction performance on static scenes, as shown in `Fig. 9`.
> >
> > ### >> **Q5 Effectiveness of using mask**
> > $ \color{red} {A5:} $ Thank you for your positive feedback on the distractor mask idea.
> > - **Qualitative comparison for mask**: We have added more qualitative comparisons in `Fig. 11`, showing VGTW's predicted masks against those from MonST3R, showing that our approach identifies distractors more precisely.
> > - **w/o DAT performance**: The mask head alone improves reconstruction quality in static regions, but it does not match the performance of the full VGTW. See Table from `A3`.
> >
> > ### >> **Q6 Training details**
> > $ \color{red} {A6:} $ Thank you for your question. Here are the details of our training setup:
> > **Computational Resources**: We trained the model using 8 GPUs with 48GB A5000 each for approximately 50 epochs, which took around 4 hours. The optimizer used was AdamW with a learning rate of 1e-5.
> > **No 3D Supervision**: We train our model without 3D supervision, as no dataset currently provides paired ground-truth masks, clean depths, and point clouds for in-the-wild scenes. Consequently, we rely only on images and masks during training. At this stage, we **cannot assess the effectiveness of 3D supervision** due to the data issue. We plan to explore this direction in future work.
> >
> > ### >> **Q7 Detailed explanation to evaluation**
> > $ \color{red} {A7:} $ Thank you for pointing it out! We will clarify the evaluation procedure in detail.
> >
> > Both NeRF-on-the-go and RobustNeRF datasets contain pairs of images from the same scene, with some images free of distractors. We use a pretrained $\pi^3$ model to reconstruct pseudo ground-truth point clouds from these non-distractor images.
> >
> > For the evaluation, we randomly sample 10 images from the dataset, which may include both distractor and non-distractor images. The model then predicts the point cloud for these images, the inference process, as detailed in `A3`. Subsequently, following the procedures of VGGT and $\pi^3$, we align the predicted point maps to the ground truth using the Umeyama algorithm for coarse Sim(3) alignment.After coarse alignment, we perform refinement using the Iterative Closest Point (ICP) algorithm.
> >
> > Finally, we evaluate the aligned predicted point clouds using three metrics: Accuracy (Acc.), Completion (Comp.), and Normal Consistency (NC.).
> >
> > ### >> **Q8 Refine tables**
> > $ \color{red}{A8:} $ Thank you for your suggestion. We will add average scores to `Tables 1–3` in the final version to facilitate easier comparisons and enhance readability.
> >
> > **`Reference`**
> > [1] Pont-Tuset, Jordi, et al. "The 2017 davis challenge on video object segmentation." arXiv preprint arXiv:1704.00675 (2017).
> > [2] Jensen, Rasmus, et al. "Large scale multi-view stereopsis evaluation." Proceedings of the IEEE conference on computer vision and pattern recognition. 2014.
> > [3] Ren, Weining, et al. "Nerf on-the-go: Exploiting uncertainty for distractor-free nerfs in the wild." Proceedings of the IEEE/CVF Conference on Computer Vision and Pattern Recognition. 2024.

---

> ### Comment · Reviewer_qMPh · 2025-11-26
> **Official Comment by Reviewer qMPh**
>
> Thanks for the authors' response. While several of my concerns have been addressed, I still have the following comments.
>
> 1. Claims regarding static assumptions
>
> Since several recent 3D reconstruction methods (e.g., π3, which the proposed method builds upon) can already handle dynamic scenes. I recommend refining the claims in the final manuscript, like the statement in abstract:
> > "…built on a restrictive assumption: the scene is entirely static with dense correspondence."
>
> 2. Relation to Easi3R
>
> Although I appreciate the explanation of the task differences and understand that the method implicitly fine-tunes attention, the core insight is closely related to Easi3R, yet no any discussion is provided  in the Problem analysis part. Easi3R shows that 3D models may assign attention to tokens that violate epipolar constraints (so this includes dynamic objects, or distractors here), and that re-weighting attention (like masking attention here) can improve reconstruction quality. I encourage the authors to include a discussion of this connection of final manuscript.
>
> 3. Evaluation without 3D supervision
>
> To evaluate the impact of lacking 3D supervision, retraining is not necessary. A simpler way is to evaluate standard 3D reconstruction metrics (e.g., using the π3 evaluation code). Since the method produces much improved confidence maps for distractor and introduces distractor masks as extra output, it would be meaningful to understand how the point map output is affected when the whole pipeline trained without 3D supervision.
>
> I still have concerns about the practicality of using a fixed 0.5 threshold to filter half the points, as the proportion of distractor regions may vary by frame and by scene.
>
> Although the concerns above prevent me from fully supporting acceptance, my other concerns have been addressed, I will consider raising my score.

---

> ### Author Response · Authors · 2025-11-28
> **Response to Reviewer qMPh (Part III)**
>
> We sincerely thank reviewer for the nice suggestions.
>
> ### >> **Q9 Claims regarding static assumptions**
> $ \color{red} {A9:} $ Thanks for the suggestion! As suggested, we have have update the maniscript on `Line 30-31` to: "but rely on a restrictive static assumption: the scenes is entire distractor‑free with perfect cross‑view geometry". This revised statement clarifies that we not only handles **static scene**, but more general and realistic settings with **imperfect cross‑view 3D geometry**.
>
> Thanks again!
>
>
> ### >> **Q10 Relation to Easi3R**
> $ \color{red} {A10:} $ We truly appreciate this comment and agree that a discussion of Easi3R is valuable. As suggested, we have added a paragraph in `Sec. 3.2, Line 204-210` to clarify our relation to Easi3R.
>
>
> While both works note that 3D models may attend to tokens violating epipolar geometry, there are fundamental differences in scope, model design, and solution strategy:
>
> - **Problem Scope**: Easi3R analyzes dynamic scenes in `Sec 3.2` on DAVIS-16 dataset, where motion naturally induces attention to non-epipolar regions. In contrast, we focuses on scenes corrupted by unrelated distractors (e.g., transient objects or background clutter) in unstructured, unordered image collections. Thus, the anlaysis in Easi3R is insufficient to generalize to our study.
> - **Different Models**: Easi3R studies DuSt3R-like, pairwise models, and restrict cross-attention to only two views at a time. In contrast, our method uses VGGT/$\pi_3$, which performs multi-view joint attention across all images. This model difference leads to different attention patterns, necessitating new analysis.
> - **Different solutions**: Easi3R applies training-free attention re-weighting; we retrain the model to learn distractor-suppressing attention end-to-end.
>
> **Experiement: Easi3R cannot do our task good**
>
> To further clarify the gap, we evaluated Easi3R on distractor-free reconstruction using unordered datasets (e.g., RobustNeRF and NeRF-on-the-go).
>
> The results have been updated in the manuscript `Fig 11`. Our method do better than Easi3R in those datasets. While Easi3R attempts to detect distractors via low inter-frame attention correlation, it often misclassifies static scene content as distractors when large viewpoint changes occur (e.g., imaging different sides of the same object). As a result, its attention masks erroneously suppress valid static regions while retaining actual distractors.
>
>
>
> ### >> **Q11 Evaluation without 3D supervision**
> $ \color{red} {A11:} $Thank you for raising this important point! As suggested, we have supplemented our evaluation on standard 3D reconstruction metrics, by comparing VGTW and π³ on the ETH3D dataset.
>
> The results are shown below, where VGTW achieves comparable performance and even slight improvements on some metrics. This demonstrates that our lightweight DAT fine-tuning effectively preserves the model's robustness in the original domain without introducing significant bias toward the target task.
>
> |         | Acc. ↓        |               | Comp. ↓      |               | NC. ↑      |               |
> |---------|------------------------|---------------|------------------------|---------------|------------------------|---------------|
> |         | Mean            | Med.     | Mean            | Med.     | Mean            | Med.     |
> |  $\pi^3$   |     0.194              |     0.131      |      0.210               |    0.128      |         0.883           |    0.969         |
> | VGTW    |       0.189            |   0.125      |       0.206             |     0.125    |        0.861            |     0.962      |
>
> ### >> **Q12 Predefined 0.5 threshold**
> $ \color{red} {A12:} $ Thank you for this question. We have supplemented `Figure.12` to show that, while the proportion of distractors varies across scenes, a keep ratio of 50%(threshold 0.5) is an appropriate threshold. It balances preserving the completeness of the target scene while effectively filtering out most distractors and background artifacts.
>
> ------
>
> Once again, thank the reviewer for the thoughtful feedback and for recognizing the potential in our work. We will incorporate the discussion in our revision.
>
> Please let me know if you need any further clarification.

---

### Official Review · Reviewer_Lh4W · 2025-10-31

**Soundness:** 3
**Presentation:** 3
**Contribution:** 2
**Rating:** 6
**Confidence:** 5

**Summary:**

The paper proposes to improve the robustness of feed-forward 3D reconstruction models to view-inconsistent distractors such as moving objects and occluders. To this end, the authors first investigate the attention mechanism under distractors: given ground-truth distractor masks, they set the attention logits of the corresponding regions to negative infinity, which markedly improves existing models when distractors are present. Building on this observation, they introduce Distractor-Aware Training (DAT) that fine-tunes attention layers via LoRA and adds distractor-suppression and cross-view consistency losses. To demonstrate effectiveness, the authors annotate a new dataset with distractor masks, and the resulting models outperform strong baselines on real scenes with distractors.

**Strengths:**

* Originality:

  * To the best of my knowledge, this paper is the first to systematically investigate distractors in attention for feed-forward 3D reconstruction and to propose a corresponding solution that directly addresses the identified failure mode.

* Quality:

  * The experiments are comprehensive, covering different types of distractors and demonstrating effectiveness across multiple strong baselines.

* Clarity:

  * The paper is well organized: it states assumptions, conducts oracle experiments with distractor masks to validate them, and then proposes a practical method informed by these findings.
  * The method is clearly introduced with sufficient technical detail to understand and reproduce.

* Significance:

  * The problem addressed is complementary to existing feed-forward 3D reconstruction efforts; by improving robustness to distractors, the approach has the potential to become a general building block for this family of models.
  * The annotated RobustNeRF-Mask dataset can further catalyze follow-up research on making feed-forward 3D reconstruction methods more robust to distractors.

**Weaknesses:**

* The notion of “distractor” for attention needs a more precise quantitative definition. Unlike NeRF-style, per-scene optimization on a single sequence, this paper targets large-scale, feed-forward transformers with attention. In such settings, dynamic objects with continuous motion can still induce high cross-view correlation and be incorporated into reconstruction. Therefore, I would recommend to quantify distractor “lifetime” (number of frames or views), apparent speed (pixels/frame or m/s under known intrinsics/poses), and spatial extent, then analyze performance as these variables change.

* From the qualitative results, improvements in static regions appear marginal. To make the gains more pronounced and measurable, I suggest: (1) also evaluating camera pose accuracy (ATE/RPE) to show benefits beyond point/depth; (2) reporting metrics on static-only regions by masking out distractor areas per frame, so improvements to clean content are not diluted by scene-wide averaging.

* The robustness gains rely on additional supervision from annotated masks, but the current dataset scale is limited; generalization needs stronger support:

  * Assess whether fine-tuning degrades performance on distractor-free data (report deltas on clean subsets).
  * Evaluate generalization to “atypical” distractors absent or rare in training (hold-out categories, cross-domain scenes), and document any regressions.

* It remains unclear why the distractor mask head is not explicitly injected into attention, akin to the “oracle” masking in Section 3. At present, correlation between distractor and static regions is reduced via feature-space objectives alone, which may struggle when distractors share similar appearance or texture with static content. Actionable: add an inference-time attention gating variant that consumes the predicted mask (soft or hard), compare against DAT-only, and analyze cases with look-alike distractors to test whether explicit gating closes the gap to the oracle.

**Questions:**

- First, please see my weaknesses above.
- Second, Figure 3 is confusing. Since the paper uses LoRA, the intra-frame and inter-frame VGGT transformer blocks should be frozen; however, in Figure 3 these blocks are labeled as trainable. Please clarify.
- Third, how does the method perform on video data with continuously persistent moving objects, e.g., the DAVIS dataset? The experiments on real dataset is important to clarify some aforementioned weaknessness.

---

> ### Author Response · Authors · 2025-11-24
> **Response to Reviewer Lh4W (Part I)**
>
> We would like to thank the reviewer for the insightful feedback and comments. We are greatly encouraged by the feedback, as the inquiry regarding integrating masks into the attention mechanism aligns with our initial concept, and we appreciate the affirmation of this direction's potential viability.
> We are more than happy to address any questions about the motivations behind our current method design.
>
>
> ### >> **Q1 Ambiguity of distractor definition**
> $ \color{red} {A1:} $ Thank you for your suggestion—a clearer definition indeed helps readers better understand our task.
> - **Definition**: We have updated `Sec.3.2` to explicitly state that distractors encompass all elements that disrupt inter-frame consistency, including dynamic objects, transient occluders, and non-rigid deformations.
> - **Quantifying distractors**: We appreciate your recommendation to quantify attributes like lifetime, apparent speed, and spatial extent. However, reliably quantifying such metrics is challenging in unordered datasets like RobustNeRF[1] and NeRF-on-the-go[2], which feature diverse distractor types (dynamic and transient objects) and lack temporal continuity or consistent view ordering.
> Following NeRF-on-the-go, we use distractor ratio for quantification, categorizing scenes into low- to high-occlusion levels; `Tables. 6-7` provide quantitative 3D reconstruction comparisons across these levels, demonstrating performance variations. We will consider further breakdowns in future work.
>
> ### >> Q2 **Evaluation on Camera pose estimation and static-only region reconstruction**
> $ \color{red} {A2:} $ Thank you for your suggestion—these experiments indeed further validate VGTW's effectiveness. We have added evaluations on NeRF-on-the-go for camera pose estimation and static-only reconstruction performance, as shown below. We incorporate these into the revised paper.
> - **Camera pose estimation**. Thank you for this suggestion—we have added camera pose evaluations in `Table.10`. Below, we summarize performance across occlusion levels, showing VGTW's overall improvements over VGGT:
>
> |         | Low Occlusion          |               |          | Medium Occlusion       |               |          | High Occlusion         |               |          |
> |---------|------------------------|---------------|----------|------------------------|---------------|----------|------------------------|---------------|----------|
> |         | RPE_trans ↓            | RPE_rot ↓     | ATE ↓    | RPE_trans ↓            | RPE_rot ↓     | ATE ↓    | RPE_trans ↓            | RPE_rot ↓     | ATE ↓    |
> | VGGT    | 9.2286                 | 105.7482      | 1.6550   | 2.4336                 | 54.0894       | 1.1660   | 5.0823                 | 60.1764       | 0.8047   |
> | VGTW    | 9.2344                 | 105.3673      | 1.6511   | 2.4305                 | 53.9070       | 1.1576   | 5.0898                 | 60.0379       | 0.7935   |
>
> - **Static-only region comparison**. Thank you for the suggestion. We have added an evaluation restricted to static-only regions. As shown below, VGTW also improves reconstruction quality on static-only regions compared to VGGT. Specifically, for a fair comparison on an unseen dataset, we further annotate the “Corner” subset of NeRF-on-the-go with distractor masks, and use these GT masks to filter out distractor areas before computing the metrics.
>
> | | Acc↓ | Comp↓ | NC↑ |
> |--------------|------------|--------|--------|
> | VGGT | 0.0585 | 0.0759 | 0.6303 |
> | VGGT+GT mask | 0.0459 | 0.0598 | 0.7281 |
> | VGTW | 0.0397 | 0.0484 | 0.7403 |
> | VGTW+GT mask | 0.0314 | 0.0448 | 0.7573 |
>
>
> ### >> Q3 **Generalizability**
> $ \color{red} {A3:} $ Thank you for your interest in the generalization ability of VGTW. We have added cross-dataset evaluations on static scenes, dynamic videos, and unordered image collections to address this concern.
> - **Static scene**: We evaluate VGTW and VGGT on the DTU dataset [3] and the static-image subset of NeRF-on-the-go [2]. As shown in `Figure.9,` VGTW achieves comparable or slightly better reconstruction quality than VGGT on these clean scenes.
> - **Robustness to “atypical” distractors**: We further evaluated VGTW’s ability to handle unseen or rare distractors through cross-dataset experiments on dynamic video sequences from the DAVIS dataset [4], with results shown in `Figure.8`. Combined with the qualitative comparisons in `Figure.7`, these findings demonstrate that VGTW robustly handles a wide variety of distractors, including both dynamic and transient objects.

---

> > ### Author Response · Authors · 2025-11-24
> > **Response to Reviewer Lh4W (Part Ⅱ)**
> >
> > ### >> **Q4 Why not use distractor mask into attention**
> > $ \color{red} {A4:} $ Thank you for the insightful question. Thank you for the suggestion of explicitly gating attention with the predicted masks. Our current design intentionally uses DAT to softly suppress distractor features instead of hard elimination them in the attention, for three reasons:
> >
> > - **Soft Suppression vs. Hard Elimination**. Oracle masking in `Sec. 3` zeroes out attention on distractor regions and can be overly aggressive near distractor–static boundaries. In practice, this often removes useful geometry and harms completeness. For example, in `Figure.2`, oracle masking causes the donut geometry to disappear and its depth to collapse onto the table, leading to an incorrect reconstruction.
> > - **Handling similar-texture regions**. Gated attention may struggle with texture ambiguity, especially when the mask is inaccurate in similar-texture areas, potentially discarding valuable information. DAT, on the other hand, leverages the pretrained VGGT backbone to better distinguish objects from similar textures.
> > - **Efficiency**. Implementing oracle-based gating would require an additional forward pass, reducing efficiency. DAT allows us to maintain single-pass inference, keeping the process as efficient as VGGT while achieving strong results.(see Table shown below).
> >
> > |         | 5 frames     | 10 frames   | 20 frames   | 80 frames  |
> > |---------|--------------|-------------|-------------|------------|
> > | VGGT    | 0.36s        | 0.65s       | 1.38s       |8.62s       |
> > | VGTW    | 0.38s        | 0.70s       | 1.45s       |8.76s       |
> >
> > ### >> **Q5 Framework figure refinement**
> > $ \color{red} {A5:} $ Thank you for pointing this out.We have updated `Figure.3`. with only LoRA trainable to ensure lightweight fine-tuning.
> >
> > ### >> **Q6 Performance on dynamic video data**
> > $ \color{red} {A6:} $ Thank you for your suggestion regarding generalization on dynamic video data. As mentioned in `A.3`, we have also tested our method on the DAVIS dataset[4]. Our results show that VGTW outperforms VGGT in effectively suppressing the confidence in dynamic objects, leading to better removal of moving objects in distractor-free reconstruction.
> >
> > **`Reference`**
> > [1] Sabour, Sara, et al. "Robustnerf: Ignoring distractors with robust losses." Proceedings of the IEEE/CVF conference on computer vision and pattern recognition. 2023.
> > [2] Ren, Weining, et al. "Nerf on-the-go: Exploiting uncertainty for distractor-free nerfs in the wild." Proceedings of the IEEE/CVF Conference on Computer Vision and Pattern Recognition. 2024.
> > [3] Jensen, Rasmus, et al. "Large scale multi-view stereopsis evaluation." Proceedings of the IEEE conference on computer vision and pattern recognition. 2014.
> > [4] Pont-Tuset, Jordi, et al. "The 2017 davis challenge on video object segmentation." arXiv preprint arXiv:1704.00675 (2017).

---

### Official Review · Reviewer_pADS · 2025-10-31

**Soundness:** 3
**Presentation:** 3
**Contribution:** 2
**Rating:** 6
**Confidence:** 4

**Summary:**

This paper introduces Visual Geometry Transformer in the Wild (VGTW), a feed-forward multi-view 3D reconstruction framework designed to handle real-world scenarios with transient distractors such as moving people or vehicles.
Built upon prior transformer-based methods like VGGT and π³, VGTW adds:

a Distractor-Aware Training (DAT) strategy that fine-tunes attention via LoRA to suppress dynamic regions;

two novel loss functions—Distractor Suppression Loss and Cross-View Consistency Loss;

an auxiliary mask prediction head for identifying dynamic regions;

a new RobustNeRF-Mask dataset constructed using SAM2 segmentation and optical-flow consistency to generate pixel-level distractor annotations.

The resulting model can directly output clean, distractor-free 3D point clouds and camera poses without requiring any 3D supervision, showing strong performance and robustness across multiple benchmarks.

**Strengths:**

Addresses a real-world gap: VGTW is the first feed-forward 3D reconstruction framework that explicitly handles transient distractors, a limitation of prior models like VGGT or DUSt3R.

Conceptually simple yet effective: By introducing DAT and LoRA-based fine-tuning, the authors enhance robustness without altering the backbone structure.

No 3D ground-truth supervision: The model only relies on 2D distractor masks, keeping the training lightweight and practical.

Solid empirical results: Experiments show consistent improvements over strong feed-forward baselines, particularly in scenes with heavy occlusions or dynamic content.

**Weaknesses:**

Dependence on mask supervision:
The method heavily relies on the new RobustNeRF-Mask dataset, where distractor masks are generated using SAM2 and optical-flow consistency, followed by partial manual refinement.
This external dependence limits generalization and raises concerns about the method’s robustness if mask quality degrades or manual curation is reduced.

Limited and potentially unfair comparisons:

Feed-forward baselines (VGGT, MASt3R, DUSt3R, etc.) do not use any distractor supervision, so performance gains might stem from additional training signals rather than methodological novelty.

The paper does not compare against optimization-based dynamic reconstruction methods (e.g., NeRF-W, WildGaussians, SpotLessSplats). While paradigms differ, such a comparison would contextualize VGTW’s efficiency–quality trade-off.

Ablation incompleteness:
While DAT and mask head ablations are shown, there is no study on varying mask quality, missing masks, or cross-domain generalization, making it unclear how resilient the model is to imperfect annotations.

**Questions:**

Could the authors provide details on the extent of manual correction performed during RobustNeRF-Mask construction? How significant was human intervention relative to SAM2 + optical flow auto-labeling?

To ensure fairness, have the authors tested a VGGT or π³ baseline trained with the same distractor masks (e.g., applying identical supervision but without DAT) to isolate the contribution of the proposed training strategy?

Have the authors compared VGTW against dynamic-object-robust NeRF or Gaussian Splatting methods (e.g., NeRF-W, WildGaussians)? Even if paradigms differ, a runtime–quality comparison would help clarify the practical advantage of feed-forward reconstruction.

How robust is VGTW to noisy or imperfect masks? Have the authors quantified how performance drops when mask quality decreases or when no masks are available for fine-tuning?

Since the model is designed for “in-the-wild” data, have the authors tested it on completely unseen domains (e.g., night scenes, handheld videos, or fast-moving dynamic scenes) to validate generalization?

---

> ### Author Response · Authors · 2025-11-24
> **Response to Reviewer pADS (Part I)**
>
> We thank the reviewer for your thoughtful and comprehensive feedback on our submission. We value the meaningful suggestions regarding dataset quality exploration, which provide significant guidance for improving robustness, and we look forward to developing better methods to scale up and create even stronger models in future work.
>
> ### >> **Q1 Performance degradation when trained on poor quality mask**
> $ \color{red}{A1:} $ Thank you for raising this important point. Exploring the reliance of Distractor-aware Training (DAT) on supervision quality is crucial for understanding its generalization capability. In response to your suggestion, we constructed a **"dirty" dataset** where 10% of the distractor masks were incorrectly labeled as static regions, making the model misinterpret dynamic objects as part of the static scene. While VGTW trained on this noisy dataset remains more resilient than VGGT, performance degradation also exists:
> - **Quantitative degradation**, shown in `Table.13`, indicates that even when trained on this noisy dataset, VGTW still outperforms VGGT in achieving distractor-free reconstruction. However, we observed a degradation in mask estimation accuracy across all metrics, as seen in `Table.13`.
> - **Qualitative degradation**, illustrated in `Fig.10`, shows increased issues such as holes in the reconstruction and discontinuities in the same object’s mask. These artifacts result in more persistent distractors in the final point cloud, confirming that the model’s performance is sensitive to mask quality.
>
> ### >> **Q2 Baselines training lacks distractor supervision**
> $ \color{red}{A2:} $ Thank you for pointing this out. However, I must clarify that feed-forward baselines (e.g., VGGT, MASt3R, and DUSt3R) cannot be directly compared in terms of performance after fine-tuning on datasets like RobustNeRF[1] and NeRF-on-the-go[2], as these baselines rely on 3D supervision. Current datasets, however, lack paired distractor images, clean depth maps, point clouds, camera poses, and inter-frame matching annotations.
>
> **Motivation for DAT**: DAT addresses this gap by allowing lightweight fine-tuning on small-scale 2D mask data alone. Our qualitative and quantitative results (`Figs. 7-9`; `Tables. 6-10`) demonstrate that VGTW generalizes well and delivers strong performance, underscoring the value of our training strategy in low-supervision settings.
>
> ### >> **Q3 Optimization-based dynamic reconstruction methods**
> $ \color{red}{A3:} $ Thank you for raising this important question. However, our approach and optimization-based dynamic reconstruction methods (e.g., NeRF-W, WildGaussians, SpotLessSplats) operate under entirely different paradigms, making direct comparisons challenging.
>
> - **Task Difference**: Our task focuses on **feed-forward reconstruction** of distractor-free point clouds, while the NeRF-based and 3DGS-based methods are designed for **novel-view synthesis**, which involves generating images from unseen viewpoints.
>
> - **Metrics Difference**: Our evaluation metrics assess point cloud **Accuracy** and **Completion**, whereas the methods you referenced primarily use metrics like **PSNR** to evaluate the quality of rendered images, which is not directly applicable to our task.
>
> In terms of efficiency, however, we expect VGTW to be faster, as it has comparable inference time to VGGT, as shown in the table below.
>
>
> |         | 5 frames     | 10 frames   | 20 frames   | 80 frames  |
> |---------|--------------|-------------|-------------|------------|
> | VGGT    | 0.36s        | 0.65s       | 1.38s       |8.62s       |
> | VGTW    | 0.38s        | 0.70s       | 1.45s       |8.76s       |
>
>
>
> ### >> **Q4 Ablation incompleteness**
> $ \color{red}{A4:} $ Thank you for pointing out the incompleteness of our ablation study and generalizability tests. We have since added the relevant experiments.
> - **Varying mask quality**: As discussed in A1, our ablation study shows that while poor mask quality reduces VGTW's accuracy, it still provides a significant improvement over the baseline.
> - **Cross-domain generalization:** We have now included tests on the DAVIS dynamic video dataset[3], as shown in `Figure.8,` where VGTW more effectively removes moving objects in reconstructions compared to the baseline. Additionally, we tested VGTW on static scenes from the DTU dataset[4] and the NeRF-on-the-go static subset[2], as shown in `Figure.9`. Our results demonstrate that VGTW maintains high-fidelity reconstruction, and in some cases, even outperforms the baseline.

---

> > ### Author Response · Authors · 2025-11-24
> > **Response to Reviewer pADS (Part Ⅱ)**
> >
> > ### >> **Q5 Manual correction in dataset annotation**
> > $ \color{red} {A5:} $ Thank you for this question, we detail the annotation method of RobustNeRF-Mask. Our dataset annotations are **entirely human-driven**, as automatic labeling methods like optical-flow-based techniques or SAM2 propagation are not suitable for annotating RobustNeRF. This is because objects in the dataset do not consistently move across frames. The dataset includes transient objects, where some objects appear in one frame but are absent in others. Therefore, we rely heavily on manual inspection to identify inconsistencies between frames and determine which regions are distractors. SAM2 is then used to generate masks for these identified distractor objects.
> >
> > ### >> **Q6 Compare to VGGT with distractor mask**
> > $ \color{red} {A6:} $ Thank you for raising this important suggestion! To quickly test this idea, we conducted an additional experiment. We applied ground-truth (GT) masks directly to remove distractors from VGGT reconstructions. This setup provides **oracle results**, simulating a model that always predicts perfect masks under mask supervision. In other words, *VGGT + GT mask* should always outperform `applying identical supervision but without DAT`.
> >
> >
> > For a fair evaluation, we conducted additional experiments on an unseen dataset, annotating the "Corner" subset of NeRF-on-the-go[2] with GT distractor masks and filtering distractor regions post-reconstruction. As shown in the table below, VGTW is always better than the oracle *VGGT + GT mask*, and of course, better than the suggested case of `applying identical supervision but without DAT`.
> >
> > | | Acc↓ | Comp↓ | NC↑ |
> > |--------------|------------|--------|--------|
> > | VGGT | 0.0585 | 0.0759 | 0.6303 |
> > | VGGT+GT mask | 0.0459 | 0.0598 | 0.7281 |
> > | VGTW | 0.0397 | 0.0484 | 0.7403 |
> > | VGTW+GT mask | 0.0314 | 0.0448 | 0.7573 |
> >
> >
> > ### >> **Q7 Generalizability**
> > $ \color{red} {A7:} $ Thank you for your question. Please refer to our response in `A4` regarding cross-domain generalization, which demonstrates that VGTW generalizes well on unseen domains, including dynamic video data and static scenes. However, we have not yet identified an appropriate dataset to specifically test night-time scenes.
> >
> > **`Reference`**
> > [1] Sabour, Sara, et al. "Robustnerf: Ignoring distractors with robust losses." Proceedings of the IEEE/CVF conference on computer vision and pattern recognition. 2023.
> > [2] Ren, Weining, et al. "Nerf on-the-go: Exploiting uncertainty for distractor-free nerfs in the wild." Proceedings of the IEEE/CVF Conference on Computer Vision and Pattern Recognition. 2024.
> > [3] Pont-Tuset, Jordi, et al. "The 2017 davis challenge on video object segmentation." arXiv preprint arXiv:1704.00675 (2017).
> > [4] Jensen, Rasmus, et al. "Large scale multi-view stereopsis evaluation." Proceedings of the IEEE conference on computer vision and pattern recognition. 2014.

---

### Official Review · Reviewer_oJJM · 2025-11-01

**Soundness:** 3
**Presentation:** 3
**Contribution:** 3
**Rating:** 6
**Confidence:** 4

**Summary:**

The paper introduces VGTW (Visual Geometry Transformer in the Wild), an end-to-end feed-forward 3D reconstruction framework for handling inconsistent multi-view images with transient distractors.
It employs a Distractor-Aware Training (DAT) strategy with two tailored losses and an auxiliary mask head for distractor suppression, trained on the proposed RobustNeRF-Mask, a dataset with pixel-level distractor annotations that enables distractor-free 3D reconstruction without 3D supervision.

**Strengths:**

1. The paper is well written, clearly structured, and easy to follow.

2. The motivation is clear, focusing on the challenge of transient distractors in real-world multi-view 3D reconstruction.

3. The technical design is reasonable, combining distractor-aware attention, consistency losses, and a mask head in a lightweight manner.

4. The experiments demonstrate consistent improvements over baseline methods, yielding cleaner and more reliable 3D reconstructions.

**Weaknesses:**

1. **Insufficient evaluation on standard benchmarks**
The method is mainly evaluated on the dataset introduced in the paper for distractor-free 3D reconstruction. It remains unclear whether the proposed approach maintains comparable performance when input images contain no distractors (i.e., fully static scenes). An additional evaluation on standard benchmarks such as DTU [1] or ETH3D [2] would help verify generalization.

2. **Unclear dataset annotation and segmentation process**
The paper (around lines 187–188) mentions that “This may include dynamic objects, occluders, and non-rigid deformations.” as part of its distractor definition, but it does not clearly explain how the corresponding motion masks or annotations are obtained. Although the paper indicates the use of SAM for segmentation, it remains unclear how complex interactions or partial deformations are handled. For example, when a standing person opens a refrigerator, is the entire refrigerator segmented as dynamic, or only the moving door? Similarly, is the whole person segmented as dynamic, or only the moving hand? Clarification is needed on whether the segmentation results were manually refined, automatically filtered, or heuristically selected to ensure labeling correctness.

3. **Lack of comparison with dynamic reconstruction baselines**
The paper omits discussion and comparison with dynamic scene reconstruction methods such as MegaSAM [3] and MonST3R [4], which process full video sequences rather than image collections. Including at least one comparison or discussion with these approaches would strengthen the empirical analysis and better clarify the positioning of the proposed method.

4. **Potential bias in evaluation setup**
The ground truth is labeled using π³ [5], which is also included as one of the baselines. This raises potential fairness concerns in the evaluation process. One possible solution is to follow PAGE-4D [6] and perform additional evaluation on an independent dataset such as DyCheck [7] to ensure unbiased comparison.

5. **Citation formatting issue**
The citation in line 125 appears to be incorrect and contains unresolved reference symbols (“;?”).

**Reference**

[1] Large-Scale Multi-View Stereopsis Evaluation.
[2] A Multi-View Stereo Benchmark with High-Resolution Images and Multi-Camera Videos.
[3] MegaSAM: Accurate, Fast, and Robust Structure and Motion from Casual Dynamic Videos.
[4] MonST3R: A Simple Approach for Estimating Geometry in the Presence of Motion.
[5] π³: Permutation-Equivariant Visual Geometry Learning.
[6] Monocular Dynamic View Synthesis: A Reality Check.

**Questions:**

1. The method based on π³ fails to improve the NC metric on the evaluation datasets. Could the authors explain the reason behind this?

2. Although the method is described as lightweight, inference time comparisons are not reported. Could the authors provide quantitative runtime results to support this claim?

3. Could the authors provide more detailed w/o experiments to isolate the effect of each proposed component and demonstrate how each contributes to the overall performance improvement?

---

> ### Author Response · Authors · 2025-11-24
> **Response to Reviewer oJJM (Part I)**
>
> We thank the reviewer for diligent and detailed assessment of our work.
> We appreciate the valuable insights provided, particularly in highlighting descriptive ambiguities in our paper and suggesting inspiring directions for further evaluating generalization across various scenarios
>
> ### >> **Q1 Performance on static scene reconstruction**
> $ \color{red}{A1:} $ Thank you for highlighting the need for evaluation on standard benchmarks. To verify this, we have supplemented evaluations on static scene reconstruction performance, including zero-shot tests on DTU[1] and non-distractor subsets from NeRF-on-the-go[2] (as shown in `Figure.9`). While VGTW is designed for in-the-wild scenes with distractors, it maintains strong performance on fully static inputs by preserving consistent features. As shown in `Figure. 9`, VGTW achieves comparable or superior results to VGGT, e.g., clearer textures in the house reconstruction.
>
> ### >> **Q2 Distractor definition & Dataset annotation**
> $ \color{red}{A2:} $ Thank you for pointing out the need for clarification on the dataset annotation process. As described in `Sec. 5.1`, RobustNeRF-Mask relies on **human judgment** to identify distractors that disrupt cross-view consistency frame by frame, followed by SAM2 to generate pixel-level masks for each detected distractor. After initial annotation, we conducted multiple rounds of verification by several annotators to avoid omissions or boundary errors, ensuring labeling correctness.
> **Specific case explanation**: for the examples mentioned, only parts that violate inter-frame consistency are annotated as distractors (e.g., the moving hand and refrigerator door). However, in image collection datasets like RobustNeRF, where distractor variations across frames are significant, entire objects are masked.
>
> ### >> **Q3 Comparison with dynamic reconstruction baseline**
> $ \color{red}{A3:} $ Thank you for the suggestion to compare with dynamic reconstruction baselines. To address this, we have supplemented a comparison with MonST3R[3] on the NeRF-on-the-go[2] dataset .
>
> **Superior generalizability**. As demonstrated in `Figure.8`, MonST3R fails to effectively remove distractors, resulting in significant misalignment in static scene reconstruction. This issue arises because MonST3R is trained on sequential dynamic video data, which struggles to exclude complex distractors like transient objects in unordered datasets, such as NeRF-on-the-go. In contrast, VGTW successfully performs distractor-free reconstruction in both dynamic video data (DAVIS dataset [4]) and datasets with transient objects (RobustNeRF [5] and NeRF-on-the-go [2]), as shown in `Figure.7` and `Figure.8`.
>
>  **Superior efficiency**. Unlike MonST3R's pairwise image processing, VGTW processes multiple images in a single feed-forward pass, achieving greater efficiency. This is evident in the table below, where VGTW maintains competitive inference times compared to VGGT:
>
>
> |         | 5 frames     | 10 frames   | 20 frames   |
> |---------|--------------|-------------|-------------|
> | MonST3R | 14.82s       | 28.94s      | 70.99s      |
> | VGGT    | 0.36s        | 0.65s       | 1.38s       |
> | VGTW    | 0.38s        | 0.70s       | 1.45s       |
>
>
>
> ### >> **Q4 $\pi^3$-based GT annotation**
> $ \color{red}{A4:} $ Thank you for suggesting an evaluation approach. However, we believe the current evaluation setup is reasonable for the following reasons：
>
> - **Effective evaluation**: The pseudo-GT used in our evaluations is generated by $\pi^3$ with paired clean images, but all evaluations are conducted on images with distractors. If there are concerns about potential bias introduced by the $\pi^3$ model, the significant improvements VGTW shows over VGGT (our baseline) further demonstrate the effectiveness of our method, as shown in the table below:
>
> |             | Low Occlusion (Avg) |          |          | Medium Occlusion (Avg) |          |          | High Occlusion (Avg) |          |          |
> |--------|-----|-----|-----|----|----------|-----|--------|-------|----------|
> |        | Acc↓       | Comp↓    | NC↑      | Acc↓      | Comp↓    | NC↑      | Acc↓       | Comp↓    | NC↑      |
> | VGGT        | 0.040                | 0.057    | 0.663    | 0.051    | 0.12     | 0.631    | 0.032   | 0.262    | 0.625    |
> | VGTW | 0.033 (18% ↑)   | 0.029 (49% ↑) | 0.695 (5% ↑) | 0.041 (20% ↑)      | 0.095 (21% ↑) | 0.724 (15% ↑) | 0.025 (22% ↑)    | 0.228 (13% ↑) | 0.693 (11% ↑) |
>
>
> - **Less Suitable Dataset for Our Task**: The DyCheck dataset, while valuable for monocular dynamic view synthesis, lacks dense ground-truth point cloud annotations and primarily focuses on monocular videos with dynamic motions rather than multi-view images transient distractors. Therefore, it is less suited for evaluating our approach compared to NeRF-on-the-go and RobustNeRF, which are more aligned with our task of handling transient distractors in multi-view 3D reconstruction.

---

> > ### Author Response · Authors · 2025-11-24
> > **Response to Reviewer oJJM (Part Ⅱ)**
> >
> > ### >> **Q5 Citation typo**
> > $ \color{red}{A5:} $ Thank you for the reminder; we have corrected the typesetting error in `line 125.`
> >
> > ### >> **Q6 NC metric analysis**
> > $ \color{red}{A6:} $ DAT (`Sec. 4.1`) prioritizes removing spurious points from transients, but can lead to a trade-off in Normal Consistency(NC) as it filters out more points overall—resulting in cleaner but sometimes sparser reconstructions and some inconsistency.
> >
> > ### >> **Q7 Efficiency quantitative comparison**
> > $ \color{red}{A7:} $ We provide quantitative runtime comparisons in `Table.14` (also referenced in `A3`. All experiments were conducted on an RTX 4090. The inference time was evaluated for inputs of 5, 10, and 20 frames.
> >
> > ### >> **Q8 Insufficient ablation**
> > $ \color{red}{A8:} $ Thank you for your question. As detailed in `Sec. 5.4` and `Tables 5, 11, 12`, we have conducted ablation studies on RobustNeRF and NeRF-on-the-go to isolate the effects of each component (e.g., the Distractor Suppression Loss $L_{supp}$, Cross-View Consistency Loss $L_{cons}$, and the mask prediction head). These ablations evaluate point map estimation using metrics like Accuracy (Acc↓), Completion (Comp↓), and Normal Consistency (NC↑) across low, medium, and high occlusion levels, as well as per-scene breakdowns in the appendix.
> >
> > To further validate whether DAT is beneficial for static region reconstruction. Specifically, for a fair comparison on an unseen dataset, we further annotate the “Corner” subset of NeRF-on-the-go with distractor masks. Subsequently we utilize ground-truth distractor masks to filter distractor regions after reconstruction and compare VGGT and VGTW's static-only reconstruction effects. As shown in the table below, VGTW improves reconstruction quality compared to VGGT + GT mask (which simply removes distractor regions), demonstrating that DAT enhances feature consistency and reduces artifacts even in static areas.
> >
> > | Method                  | Acc↓    | Comp↓   | NC↑     |
> > |-------------------------|---------|---------|---------|
> > | VGGT                    | 0.0585 | 0.0759 | 0.6303 |
> > | VGGT + GT mask          | 0.0459 | 0.0598 | 0.7281 |
> > | VGTW                    | 0.0397 | 0.0484 | 0.7403 |
> > | VGTW + GT mask          | 0.0314 | 0.0448 | 0.7573 |
> >
> > If you would like more detailed ablation studies or experiments on specific components, we would be happy to provide additional results.
> >
> > **`Reference`**
> > [1] Jensen, Rasmus, et al. "Large scale multi-view stereopsis evaluation." Proceedings of the IEEE conference on computer vision and pattern recognition. 2014.
> > [2] Ren, Weining, et al. "Nerf on-the-go: Exploiting uncertainty for distractor-free nerfs in the wild." Proceedings of the IEEE/CVF Conference on Computer Vision and Pattern Recognition. 2024.
> > [3] Zhang, Junyi, et al. "Monst3r: A simple approach for estimating geometry in the presence of motion." arXiv preprint arXiv:2410.03825 (2024).
> > [4] Pont-Tuset, Jordi, et al. "The 2017 davis challenge on video object segmentation." arXiv preprint arXiv:1704.00675 (2017).
> > [5] Sabour, Sara, et al. "Robustnerf: Ignoring distractors with robust losses." Proceedings of the IEEE/CVF conference on computer vision and pattern recognition. 2023.

---

> > > ### Comment · Reviewer_oJJM · 2025-11-27
> > >
> > > Thank you for your detailed response, which addresses most of my concerns. The inclusion of quantitative results on static benchmarks would further strengthen the evaluation. After reviewing the rebuttal and other reviews, I maintain my original rating.

---

> > > > ### Author Response · Authors · 2025-11-28
> > > > **Response to Reviewer oJJM (Part III)**
> > > >
> > > > ### >> **Q9 More quantitative results on static benchmarks**
> > > > $ \color{red} {A9:} $Thank you for this suggestion! As suggested, we have supplemented our quantitative evaluation on static reconstruction benchmark, by comparing VGTW and π³ on the ETH3D dataset.
> > > >
> > > > The results are shown below, where VGTW achieves comparable performance and even slight improvements on some metrics. This demonstrates that our lightweight DAT fine-tuning effectively preserves the model's robustness in the original domain without introducing significant bias toward the target task.
> > > >
> > > > |         | Acc. ↓        |               | Comp. ↓      |               | NC. ↑      |               |
> > > > |---------|------------------------|---------------|------------------------|---------------|------------------------|---------------|
> > > > |         | Mean            | Med.     | Mean            | Med.     | Mean            | Med.     |
> > > > |  $\pi^3$   |     0.194              |     0.131      |      0.210               |    0.128      |         0.883           |    0.969         |
> > > > | VGTW    |       0.189            |   0.125      |       0.206             |     0.125    |        0.861            |     0.962      |

---

### Official Review · Reviewer_qYpT · 2025-11-02

**Soundness:** 3
**Presentation:** 3
**Contribution:** 3
**Rating:** 6
**Confidence:** 4

**Summary:**

The paper introduces a fine-tuning approach based on the state-of-the-art geometric foundation models (e.g., VGGT), to tackle the distrator regions without point correspondence across multi-view input. Author first demonstrate the visualization on the attention map of original inference between using the distrator mask and w/o the mask. Then a couple of loss supervisions are proposed to enhance the feature similarty on the real matched points while suppressing the distractors. Besides, a dataset containing distractor mask is built and released for better problem setup. Extensive experiments showcase the effectiveness of the method.

**Strengths:**

(1) The paper is well motivated and handling the non-matched regions is an important yet challenging problem in multi-view geometry in a long run. The visualization of the cross attention map for those regions are admirable, to make readers better elaborate the challenge.

(2) The introduced loss functions are techncially sound to enhance the feature similarity on the true positive matched regions, while suppressing the false positive ones. The loss functions are simple yet effective to be designed and implemented.

(3) The experiments have demonstrated the validness of the design.

**Weaknesses:**

(1) The scale and diversity of constructed dataset with the distractor mask is limited (1000 annotated images on based on a single RobustNeRF) dataset, making the problem hard to be scaled up and extended. I was wondering whether using some pretrained optical flow network or dense point matching network, or simply SAM2, can scale up the annotation dataset effectively. By training on diverse dataset, the proposed method could demonstrate the generalizability over wild images.

(2) The method is only evaluated on NeRF-on-the-go dataset, which is hard to measure the generalizability and zero-shot capacity of the proposed method. Besides, authors train and evaluate the framework on the same dataset (RobustNeRF), which makes the technical contribution less convincing since all the baseline method are not trained on this dataset. I would suggest authors to do multiple cross-dataset evaluation to validate the generalizability of the method.

**Questions:**

Overall the motivation of the paper is encouraging and the technical contribution is clear. However, I still have concerns on the scalability and evaluation protocol as presented in the 'weakness' part. I hope authors could supplement more experiments to further demonstrate the generlizability of the method.

---

> ### Author Response · Authors · 2025-11-24
> **Response to Reviewer qYpT**
>
> We thank the reviewer for the constructive comments and would like to address them as follows. We appreciate the insightful suggestions on scaling up.
>
> ### >> **Q1 Dataset annotation & Scalability**
> $ \color{red}{A1:} $ Thank you for raising the important issue of dataset annotation and scalability. However, annotating transient distractors remains challenging and difficult to scale up effectively.
>
> - **Dataset Annotation**: Currently, the annotation process is entirely human-driven. We rely on manual inspection to identify distractors frame by frame. This is necessary because datasets like RobustNeRF[1] and NeRF-on-the-go[2] consist of unordered image collections, and the types of distractors are complex. They include dynamic objects and transient objects, some of which only appear in a single frame and are absent in others. Due to this variability, optical flow-based methods or SAM2 propagation cannot effectively automate the labeling of these distractors.
> - **Training for Enhanced Generalization**: We agree that scaling up the dataset can enhance the model's generalization. Training on larger datasets generally improves performance across different domains. However, our work demonstrates that with Distractor-aware Training (DAT), even with lightweight fine-tuning on a small-scale dataset, we can achieve strong cross-dataset generalizability. In the future, we aim to train a more powerful model on larger datasets to further improve its generalization capacity.
>
>
>
> ### >> **Q2 Generalizability**
> $ \color{red} {A2:} $
> Thank you for raising this important issue. We have added **cross-dataset evaluations** to further validate the **generalizability** of our method.
>
> - **Generalization to Unordered Images**: First, the zero-shot evaluation on NeRF-on-the-go is a reasonable test of generalization. Both NeRF-on-the-go and RobustNeRF are unordered image datasets containing transient and dynamic distractors, making them closely aligned with our task. Since NeRF-on-the-go is a completely unseen dataset for VGTW, our evaluation shows that VGTW can successfully reconstruct high-fidelity, distractor-free scenes from unordered images with distractors, as shown in `Figure.7`.
>
> - **Generalization to Dynamic Video**: We also performed zero-shot testing on the DAVIS[3] dynamic video dataset, where the challenge is to detect and filter out moving objects in scenes with persistent motion. The results, shown in `Figure.8`, demonstrate that VGTW effectively identifies and removes moving objects, such as the walking camel, while preserving static objects, like the stationary camel. In comparison, the baseline methods fail to remove these dynamic objects, resulting in motion "ghosting."
>
> - **Generalization to Static Scenes**: Additionally, we evaluated VGTW on static scenes from DTU dataset[4] and non-distractor subsets from NeRF-on-the-go[2]. We found that it maintains comparable high-fidelity reconstruction performance to the baseline, as shown in `Figure.9`. In some cases, such as the reconstruction of building textures, VGTW even outperforms the baseline.
>
> **`Reference`**
> [1] Sabour, Sara, et al. "Robustnerf: Ignoring distractors with robust losses." Proceedings of the IEEE/CVF conference on computer vision and pattern recognition. 2023.
> [2] Ren, Weining, et al. "Nerf on-the-go: Exploiting uncertainty for distractor-free nerfs in the wild." Proceedings of the IEEE/CVF Conference on Computer Vision and Pattern Recognition. 2024.
> [3] Pont-Tuset, Jordi, et al. "The 2017 davis challenge on video object segmentation." arXiv preprint arXiv:1704.00675 (2017).
> [4] Jensen, Rasmus, et al. "Large scale multi-view stereopsis evaluation." Proceedings of the IEEE conference on computer vision and pattern recognition. 2014.

---

> > ### Comment · Reviewer_qYpT · 2025-11-26
> >
> > The comments from authors mostly address my concerns. I will keep my current ratings unchanged.

---

> > > ### Author Response · Authors · 2025-11-26
> > >
> > > Dear reviewer qYpT, we appreciate your response. Should you have any questions, please feel free to reach out anytime. We're always here and happy to respond.

---

### Public Comment · ~Yanqi_Bao1 · 2025-11-13
**An additional discussion**

HI, thank you for the excellent work on VGGT! I would like to respectfully suggest that "Distractor-free Generalizable 3D Gaussian Splatting" (https://arxiv.org/abs/2411.17605) might be worth mentioning in Sec. 2.1, as this work similarly focuses on feed-forward distractor-free 3DGS reconstruction and was released last November. I hope this suggestion is helpful.

---

### Author Response · Authors · 2025-11-25
**Response to All Reviewers**

We sincerely thank all reviewers for their constructive feedback. We deeply appreciate the following positive comments:
- **Novel task**: `pADS`, `Lh4W`
- **Clear motivation analysis**: `qYpT`,`oJJM`, `qMPh`
- **Simple but effective design**: `qYpT`, `oJJM`, `pADS`
- **Effective exprimental validation**: `qYpT`, `oJJM`, `pADS`, `Lh4W`, `qMPh`
- **Dataset contribution**: `Lh4W`

We wil address the specific questions and concerns raised by the reviewers in the subsequent sections of this rebuttal.

---

### Comment · Area_Chair_fACJ · 2025-11-27

Dear Reviewers,

Thank you for your thoughtful evaluations of this submission. The authors have provided their responses and clarifications during the discussion phase. To ensure a well-informed final decision, I kindly encourage you to continue the discussion by reviewing the authors’ replies and adding any follow-up thoughts you may have.

If any of your original concerns remain unresolved, please feel free to raise them directly in the discussion thread.

Thank you again for your time and valuable contributions.

Best regards,
Area Chair

---

### Author Response · Authors · 2025-12-01
**Reponse to new Area Chair**

Dear Area Chair,

Thank your time and effort in managing the review process for our paper. We understand that this requires significant additional effort, and we are truly grateful for your time and support. To facilitate your review, we would like to provide a brief summary of the discussion period.

**In the pre-discussion state**, It is very encouraging that **4/5 reviewers support acceptance with high confidence**. The reviewers provided positive feedback on several key aspects:
- Novel task: `pADS`, `Lh4W`
- Clear motivation analysis: `qYpT`,`oJJM`, `qMPh`
- Simple but effective design: `qYpT`, `oJJM`, `pADS`
- Effective exprimental validation: `qYpT`, `oJJM`, `pADS`, `Lh4W`, `qMPh`
- Dataset contribution: `Lh4W`

**During the discussion phase**, we actively addressed all constructive questions and suggestions raised by the reviewers.
We are pleased to report that: Reviewer `qMPh` **has promised to increase score** and is likely to **fully support acceptance** if our new responses address his new concerns.

-  **Please note**: This promission of rating increase occurred **prior to the wide OpenReview anonymity breach**.
- Reviewers `qYpT` and `oJJM` are satisfied that our responses addressed most of their concerns.

**Response to common concern/suggestion:**

We appreciate the efforts of all five reviewers (`qYpT`, `oJJM`, `pADS`, `Lh4W`, `qMPh`) and their valuable suggestions. Here is a summary of key shared concerns and our responses:

- **Generalizability** (`qYpT`, `pADS`, `Lh4W`): We further demonstrate that VGTW exhibits strong generalizability to cross-domain scenarios, including *dynamic video* and *static scenes*. Specifically, we added qualitative and quantitative experiments, shown in `Figures.8–9` and tables in responses `A3` and `A11` to reviewer `qMPh` to show it's ability to reconstruct high-fidelity distractor-free points clouds.

- **Data Annotation** (`qYpT`, `oJJM`, `pADS`): We added a detailed illustration of the annotation process and further analyzed the impact of dataset quality on model performance. The description of the annotation process is shown in `Sec. D of the Appendix`, and the quality analysis is supported by qualitative and quantitative results in `Figure.10` and `Table.13`.
- **Comparison to Dynamic Reconstruction Models** (`oJJM`, `qMPh`): Dynamic reconstruction models (e.g., Easi3R and MonST3R) are fundamentally different from our VGTW. We analyzed the differences in terms of *task scope*, *paradigm*, and *model design* in response `A10` to reviewer `qMPh`. Additional qualitative and quantitative results demonstrating their unsuitability for our task are shown in `Figure.11` and `Table.14`.

**We have incorporated the necessary changes into the revised manuscript**. All modifications are highlighted in *pink* for ease of reference.

Finaly, we sincerely thank you for your extra effort in managing the recent OpenReview emergency. We deeply appreciate your dedication to ensuring a fair review process during this challenging time.
Sincerely,
The Authors

---

### Meta-Review · Area_Chair_518Y · 2025-12-28

**Summary:**

This paper presents Visual Geometry Transformer in the Wild, which is an end-to-end framework for robust reconstruction from inconsistent views.

The paper involves some main concerns for example, overlooking the citation and discussion of recent similar research works, overstated claims, limited evaluation, limited comparisons, annotation clarity etc. The method is based on a simple design to suppress distractor regions in multi-view 3D reconstruction. Authors had to use their introduced RobustNeRF-Mask dataset which was however noted not to be sufficiently large. This resulted in a mixed performance where sometimes VGTW_VGGT wins, sometimes VGTW_\pi^3, sometimes pi^3, and sometimes DUST3R. More work is required to obtain clear cut performance.

**Reviewer Concerns:**

Reviewer qMPh was concerned that that "The paper repeatedly suggests that VGGT/π3 conceptually cannot handle dynamic scenes" which requires addressing.

Rev. Lh4W notes that "The robustness gains rely on additional supervision from annotated masks, but the current dataset scale is limited" also remains unresolved.

**Reviewer Scores:**

Very likely concerns of Reviewers qMPh and Lh4W would have persisted.

---

### Decision · Program_Chairs · 2026-01-26

Reject